# ACCELERATING NEURAL NETWORK TRAINING: AN ANALYSIS OF THE ALGOPERF COMPETITION

**Priya Kasimbeg**[1]*  **Frank Schneider**[2]*  **Runa Eschenhagen**[3]  **Juhan Bae**[4,5]
**Chandramouli Shama Sastry**[4,6]  **Mark Saroufim**[7]  **Boyuan Feng**[7]  **Less Wright**[7]
**Edward Z. Yang**[7]  **Zachary Nado**[1]  **Sourabh Medapati**[1]
**Philipp Hennig**[2]  **Mike Rabbat**[7]  **George E. Dahl**[1]†
[1]Google DeepMind  [2]University of Tübingen  [3]University of Cambridge
[4]Vector Institute  [5]University of Toronto  [6]Dalhousie University  [7]Meta

## ABSTRACT

The goal of the ALGOPERF: TRAINING ALGORITHMS competition is to evaluate practical speed-ups in neural network training achieved solely by improving the underlying training algorithms. In the external tuning ruleset, submissions must provide workload-agnostic hyperparameter search spaces, while in the self-tuning ruleset they must be completely hyperparameter-free. In both rulesets, submissions are compared on time-to-result across multiple deep learning workloads, training on fixed hardware. This paper presents the inaugural ALGOPERF competition's results, which drew 18 diverse submissions from 10 teams. Our investigation reveals several key findings: (1) The winning submission in the external tuning ruleset, using DISTRIBUTED SHAMPOO, demonstrates the effectiveness of non-diagonal preconditioning over popular methods like ADAM, even when compared on wall-clock runtime. (2) The winning submission in the self-tuning ruleset, based on the SCHEDULE FREE ADAMW algorithm, demonstrates a new level of effectiveness for completely hyperparameter-free training algorithms. (3) The top-scoring submissions were surprisingly robust to workload changes. We also discuss the engineering challenges encountered in ensuring a fair comparison between different training algorithms. These results highlight both the significant progress so far, and the considerable room for further improvements.

## 1 INTRODUCTION

Deep neural networks are powerful models that excel in tasks such as image recognition, natural language processing, and speech recognition. However, training these models often requires significant computational resources as well as careful (and sometimes brittle) training recipes, including meticulous hyperparameter tuning. A big practical issue in the usability of *training algorithms*, for instance, is that many choices are still left to the practitioner. How should the learning rate be tuned? In what range? Using what schedule? These are very crucial decisions that can make or break the training process and, critically, determine which methods perform best. Despite training algorithms being such a fundamental part of the deep learning pipeline, the community has been unable to identify which training methods are the state of the art. Previous empirical comparisons have suffered from a number of issues, e.g., weak baselines, not fully accounting for hyperparameter tuning, or failing to properly control for potential confounding factors like model architecture changes. Dahl et al. (2023) proposed the ALGOPERF: TRAINING ALGORITHMS benchmark (Section 2) to measure speed-ups in neural net training due to algorithmic improvements, while addressing the aforementioned issues. This competitive, time-to-result benchmark uses multiple realistic deep learning workloads on fixed hardware, allowing submitters to innovate solely on the training algorithms. It allows submissions under two different hyperparameter tuning rulesets: an *external tuning ruleset* that scores submissions based on the best result from a handful of hyperparameter configurations, and a *self-tuning ruleset* without hyperparameters that counts any time a submission spends tuning as part of the training time.

---

*Equal contributions.
†Corresponding author gdahl@google.com.

In this paper, we present the results of the inaugural ALGOPERF: TRAINING ALGORITHMS competition, a competition open to the entire machine learning community, based on the ALGOPERF benchmark proposed in Dahl et al. (2023). Our analysis of $\approx 4000$ training runs reveals several key findings on accelerating neural net training through improved training algorithms:

- The winning submission in the external tuning ruleset, based on DISTRIBUTED SHAMPOO (Anil et al., 2020; Shi et al., 2023), demonstrates that non-diagonal preconditioning methods can outperform currently popular diagonal methods, such as ADAM (Kingma & Ba, 2015), in terms of wall-clock training time. Across eight deep learning workloads, this submission achieved on average a $\approx 28\%$ faster model training compared to the baseline, which used NADAMW (Dozat, 2016; Loshchilov & Hutter, 2019) (Section 3).
- In the self-tuning ruleset, the winning submission based on SCHEDULE FREE ADAMW (Defazio et al., 2024) was the only entry surpassing the baseline, providing $\approx 8\%$ faster average training. It was also $\approx 10\%$ faster than the *external* tuning baseline across the seven base benchmark workloads both algorithms trained successfully, without any hyperparameters or parallel tuning (Section 3).[1] This result establishes a new state-of-the-art for hyperparameter-free training algorithms and highlights the exciting potential of fully automated neural network training.
- The top-scoring submissions are characterized by their consistent performance across workloads, including robustness to minor workload modifications, e.g. changes to activation functions or normalization layers (Table 3). This suggests that, at least in a competitive evaluation context, achieving consistent performance across workloads is a major challenge for training algorithms operating under restricted runtime and tuning budgets.

Building the training harness and software for the ALGOPERF: TRAINING ALGORITHMS competition to enable fair and meaningful comparisons between training algorithms, especially across the deep learning frameworks JAX (Bradbury et al., 2018) and PYTORCH (Paszke et al., 2019), required substantial engineering effort. Section 4 discusses these engineering challenges, while Section 5 highlights lessons learned as well as opportunities to improve future iterations of the benchmark.

## 2 SUMMARY OF BENCHMARKING METHODOLOGY

The ALGOPERF: TRAINING ALGORITHMS benchmark evaluates the effectiveness of training algorithms by measuring how quickly they can achieve specific evaluation metric goal values across various realistic deep learning workloads. These per-workload, time-to-result measurements are performed on a fixed hardware configuration, and account for all required workload-specific tuning. The final benchmark score aggregates across workloads to identify more efficient general-purpose training algorithms for deep learning. Below, we briefly summarize the ALGOPERF benchmark by explaining key terms. For a more detailed discussion of the benchmark's motivation, description, and justification see Dahl et al. (2023) and the competition rules[2] & documentation[3]. For the ALGOPERF competition, we solicited submissions from the entire machine learning community, and made several modifications to the benchmark as it was initially proposed by Dahl et al. (2023), which are summarized in Appendix A.2.

**Workloads.** The benchmark features multiple neural network training tasks, called *workloads*, each consisting of a dataset, model, loss function, target metric, validation target and runtime budget. The submissions' objective is to train these workloads as quickly as possible; if the target is not reached within the runtime budget, the run receives an infinite score. Designed to reflect real-world deep learning training scenarios, the workloads cover several key domains. The benchmark includes two types of workloads: *fixed base workloads* (Table 2) directly affect the benchmark score, and *held-out workload variants* (Table 3) discourage submissions to overfit the benchmark's fixed workloads and ensure robustness to natural workload changes. While held-out workloads do not contribute directly to the benchmark score, failure to train a held-out workload quickly enough will invalidate the submission's score for the corresponding fixed base workload.

**Submissions.** Submitted training algorithms must adhere to the fixed ALGOPERF API (Dahl et al., 2023, Sec. 4.2) and are limited to four submission functions: (1) `update_params` is responsible for

---

[1]Note, the two speed-ups for SCHEDULE FREE ADAMW are computed across different sets of workloads.
[2]`github.com/mlcommons/algorithmic-efficiency/[...]/COMPETITION_RULES.md`
[3]`github.com/mlcommons/algorithmic-efficiency/[...]/DOCUMENTATION.md`

modifying the network's parameters during training and typically involves optimization algorithms, such as SGD, ADAM, or a custom method. (2) `init_optimizer_state` allows the creating of the training algorithm's internal state, e.g. to define learning rate schedules. (3) `data_selection` allows control of how batches are constructed, to use techniques such as curriculum learning or data echoing (Choi et al., 2020). (4) `get_batch_size` defines a batch size for each workload, e.g., participants can predetermine the largest batch size fitting in the competition hardware's memory. For the external tuning ruleset (see below), participants can also provide a workload-agnostic search space for hyperparameters. This limited API isolates training speed-ups resulting from improvements to the training algorithm itself, rather than pipeline optimizations or model changes. The API also ensures that submissions can be applied to generic deep learning workloads by being fully specified without any workload-specific behavior (apart from the batch size).

**Tuning rulesets.** Submissions compete under two distinct rulesets governing hyperparameter tuning. The *external tuning ruleset* simulates hyperparameter tuning with limited parallel resources. In this ruleset, hyperparameters are tuned using five independent *trials*, with hyperparameter configurations sampled via quasirandom search (Bousquet et al., 2017) from the submission's defined search space,[4] and are scored based on the runtime of the trial that achieves the validation target the fastest. To produce a more consistent final score, the tuning process is repeated five times across five different tuning *studies*, where each study receives a different random seed for the workload's model and data initialization. The final *workload score* used in the benchmark's scoring procedure is the median of the best training time from each of the five studies. The *self-tuning ruleset* simulates fully automated hyperparameter tuning during training on a single machine. This includes submissions that use the same hyperparameters across all workloads (e.g. ADAMW with defaults for all hyperparameters including regularization) or those that perform inner-loop tuning during the training run. Anticipating that they may require more time to reach the target, self-tuning submissions have three times the runtime budget of external-tuning submissions (see Table 2). Similar to the external tuning ruleset, five studies are conducted (although each self-tuning study consists of a single trial), and the median runtime across all studies determines the workload score.

**Benchmark score.** A submission's *benchmark score* is based on its individual workload scores relative to those of other submissions, aggregated using performance profiles (Dolan & Moré, 2002; Dahl et al., 2023, Sec. 4.5). A performance profile plots the fraction of workloads where a submission trains successfully (achieves the target performance) within a factor of $\tau$ of the time required by the per-workload fastest submission, for different values of $1 \leqslant \tau \leqslant \tau_{\max}$ (see Figures 1 and 3). For instance, the height of submission $s$'s performance profile at $\tau = 1.5$ is the fraction of workloads where $s$ trains successfully and within $1.5\times$ the wall-clock time the best competitor requires on that workload. The final scalar *benchmark scores* (Figures 1a and 1c) are the normalized area under the performance profiles, with 1.0 corresponding to a submission that was faster than all competitors on all workloads.

**Computational costs.** To score all competition submissions, we conducted 3850 and 420 runs in the external tuning and self-tuning ruleset respectively. On average, scoring an external submission required $\approx 3469$ hours, and $\approx 1847$ hours for a self-tuning submission, totaling $\approx 49{,}240$ hours on the competition hardware ($8\times$NVIDIA V100 GPUs) (see Appendix A.3).

## 3 RESULTS

The two ALGOPERF: TRAINING ALGORITHMS competition leaderboards (Figure 1) rank submissions in each tuning ruleset by their benchmark score.[5] Although five submissions outperformed the baseline (Table 4) in the external tuning ruleset, only the winning submission was competitive in the self-tuning ruleset. The remaining self-tuning submissions (Table 5) scored significantly lower than the self-tuning baseline, highlighting the challenge of fully automatic training algorithms that must pay the training time cost of any workload-specific hyperparameter tuning. Although there is at least one submission that could successfully train each workload, no submission reached the

---

[4]Submissions can alternatively provide a list of five hyperparameter configurations that will be sampled without replacement.

[5]Note that the external-tuning baseline is different from the self-tuning baseline; each was tuned and specialized specifically for its ruleset. Similarly, there are two distinct SCHEDULE FREE ADAMW submissions, one for each ruleset. We specify which one we are referring to if the context does not make it clear.

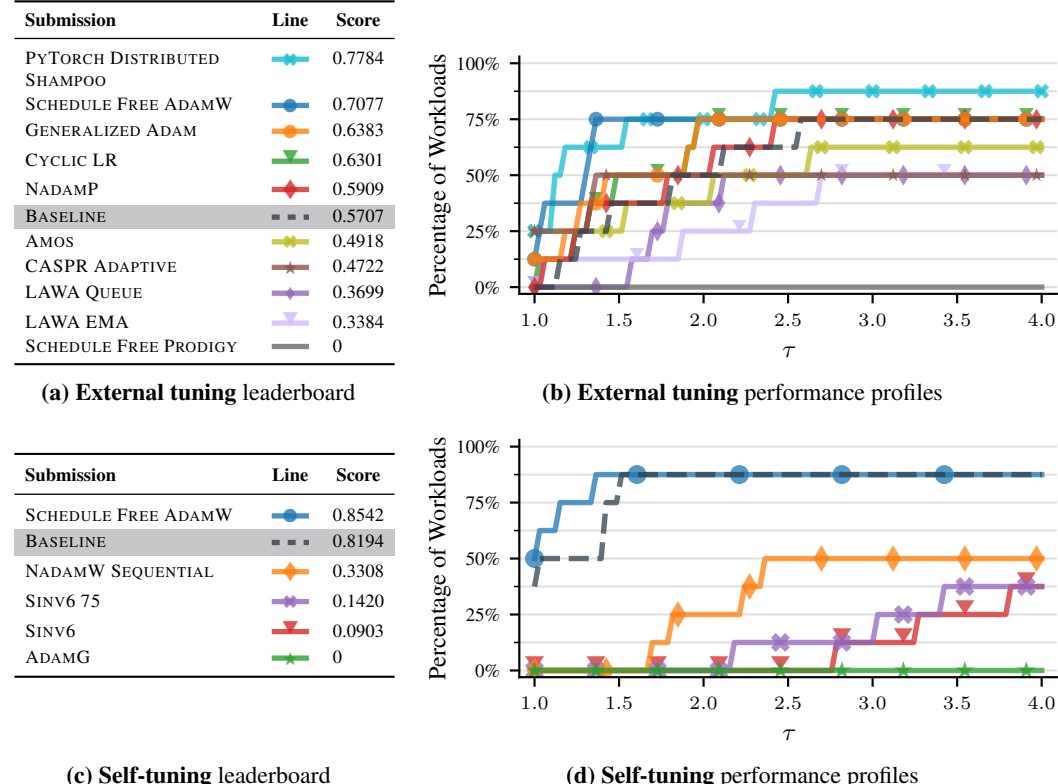

**Figure 1: ALGOPERF competition leaderboard & performance profiles** for all submissions to the external (*top*) and self-tuning (*bottom*) ruleset. The leaderboards (a, c) are ranked by the submissions' benchmark scores, rounded to four significant digits. Higher scores indicate faster training. Note, scores are *not* comparable between rulesets. In the performance profiles (b, d), each line represents a submission. A step at $\tau$ indicates that, for one workload, this submission reaches the target within $\tau$ times the runtime of the fastest submission for that workload and ruleset.

target on every workload (Figure 1; see Section 3.1 for more details). This result simultaneously demonstrates the benchmark's feasibility and its difficulty as well as the significant potential for future improvement in training algorithms. There were strong submissions in both PYTORCH & JAX, suggesting that, within the benchmark's codebase, workload implementations in both frameworks are, at least to some degree, sufficiently similar in speed and memory usage (see Section 4).

## 3.1 PER-WORKLOAD RUNTIMES

Table 1 provides a detailed overview of the submissions' runtimes across all workloads, normalized by the *external tuning runtime budget*. Workloads with runtimes deemed infinite for scoring purposes are marked in gray, with corresponding symbols explaining the reason (see the caption of Table 1). The winning submissions in both rulesets excelled by reliably training most workloads rather than being the fastest on every one. For example, PYTORCH DISTRIBUTED SHAMPOO was the fastest on "only" 2 of the 8 base workloads, while SCHEDULE FREE ADAMW led in 4 of 8 in the self-tuning ruleset. We also observed substantial variations in workload runtimes, even among competitive methods. PYTORCH DISTRIBUTED SHAMPOO, for instance, took over twice as long on WMT as the fastest submission. This indicates that there is still ample potential to improve training algorithms. In the self-tuning ruleset, *if* the winning submission successfully trained a workload, it did so within the *external* tuning runtime budget. This suggests that future benchmarks could significantly reduce the self-tuning runtime budget to save computational costs, perhaps matching the external tuning one.

**RESNET workload.** The RESNET workload is notable, since only one submission, GENERALIZED ADAM, could *reliably* train it to the target performance within the given budget. This may be

**Table 1: Normalized submission runtimes across all workloads.** All runtimes are normalized using the *external tuning* runtime budget, with the fastest submission per workload in each ruleset highlighted in bold. Workload runtimes considered infinite for scoring are marked in gray, with a (suffix) symbol explaining the reason. `inf` denotes that a submission did not reach the workload target within the allowed runtime budget. `NaN` indicates an error (such as running out of memory) before any evaluation occurred. A † indicates that a held-out score is ignored due to the submission not reaching the target on the corresponding base workload, while a ‡ indicates that a base workload score is ignored because the submission did not successfully train the associated held-out workload. Runs that are not within $4\times$ the fastest (valid) workload runtime are marked with ∗.

**(a) External tuning ruleset**

| | CRITEO 1TB | | FASTMRI | | RESNET | | VIT | CONFORMER | | DEEP SPEECH | OGBG | | WMT | |
|---|---|---|---|---|---|---|---|---|---|---|---|---|---|---|
| | Base | H.O. | Base | H.O. | Base | H.O. | Base | Base | H.O. | Base | Base | H.O. | Base | H.O. |
| AMOS | inf | inf | 0.33 | 0.49 | inf | **0.55**† | 0.65 | **0.71** | 0.57 | **0.57** | 0.60* | 0.89* | 0.68 | 0.37 |
| BASELINE | 0.94 | 0.08 | 0.23 | 0.51 | inf | 0.94† | 0.91 | 0.90 | 0.83 | 0.65 | 0.42‡ | 0.68* | 0.86 | 0.35 |
| CASPR ADAPTIVE | NaN | NaN | **0.13** | **0.15** | inf | inf | 0.58 | inf | 0.59† | 0.75 | **0.12** | **0.12** | 0.67‡ | NaN |
| CYCLIC LR | 0.67 | 0.08 | 0.25 | 0.44 | inf | inf | 0.81 | 0.94 | 0.92 | 0.70 | 0.38‡ | 0.51* | 0.49 | 0.35 |
| GENERALIZED ADAM | 0.83 | 0.05 | 0.18 | 0.39 | **0.97** | 0.88 | 0.84 | inf | 0.83† | 0.68 | 0.31‡ | 0.64* | 0.63 | 0.33 |
| LAWA EMA | 0.69 | 0.09 | 0.29 | 0.57 | inf | inf | 0.80 | inf | inf | inf | 0.57* | 0.73* | 0.89 | 0.39 |
| LAWA QUEUE | inf | 0.14† | 0.22 | 0.55 | inf | inf | 0.66 | inf | inf | inf | 0.25 | 0.24 | 0.56 | 0.22 |
| NADAMP | 0.80 | 0.07 | 0.22 | 0.49 | inf | 0.90† | 0.88 | 0.94 | 0.85 | 0.60 | 0.43‡ | 0.74* | 0.80 | 0.47 |
| SCHEDULE FREE ADAMW | 0.67 | 0.05 | 0.13 | 0.41 | inf | inf | 0.57 | 0.92 | **0.57** | 0.78 | 0.29‡ | 0.61* | **0.33** | **0.12** |
| SCHEDULE FREE PRODIGY | NaN | NaN | 0.21‡ | 0.65* | inf | inf | inf | inf | inf | inf | 0.61* | inf | inf | 0.40† |
| PYTORCH DISTR. SHAMPOO | **0.65** | **0.03** | 0.15 | 0.22 | inf | 0.93† | **0.43** | 0.78 | 0.68 | 0.62 | 0.18 | 0.19 | 0.80 | 0.25 |

**(b) Self-tuning ruleset**

| | CRITEO 1TB | | FASTMRI | | RESNET | | VIT | CONFORMER | | DEEP SPEECH | OGBG | | WMT | |
|---|---|---|---|---|---|---|---|---|---|---|---|---|---|---|
| | Base | H.O. | Base | H.O. | Base | H.O. | Base | Base | H.O. | Base | Base | H.O. | Base | H.O. |
| ADAMG | inf | inf | inf | inf | inf | inf | inf | inf | inf | inf | inf | inf | inf | inf |
| BASELINE | 0.75 | **0.07** | 0.22 | 0.51 | inf | inf | 0.95 | **0.94** | 0.92 | **0.65** | 0.46 | 0.69 | **0.84** | 0.59 |
| NADAMW SEQUENTIAL | 2.96‡ | 0.57* | 0.27 | **0.44** | inf | inf | 1.58 | inf | 1.16† | 1.45 | 0.55 | 0.96 | 2.36‡ | 1.57* |
| SCHEDULE FREE ADAMW | **0.75** | 0.25 | **0.15** | 0.58 | inf | inf | **0.68** | 0.97 | **0.61** | 0.88 | **0.32** | **0.56** | 0.94 | **0.21** |
| SINV6 | NaN | NaN | 0.49 | 0.87 | inf | inf | inf | inf | 1.82† | 2.47 | 1.35* | inf | 2.32 | 0.46 |
| SINV6 75 | NaN | NaN | 0.45 | 0.80 | inf | inf | inf | inf | 2.55† | 2.21 | 1.50* | inf | 1.82 | 0.44 |

surprising, as the RESNET target threshold was derived using the same procedure as all other workloads. The hyperparameter search space used for target-setting (see Dahl et al., 2023) may have been more suitable for this workload given its well-studied nature. Although only one submission achieved a finite RESNET workload score (recall a finite workload score requires at least 3 out of the 5 repetition studies to train successfully), several others met the target in at least one study. NADAMP, PYTORCH DISTR. SHAMPOO, and the BASELINE hit the target at least once but were not consistent, with the remaining studies falling just short (Figure 2). These results support using the median across multiple studies to determine a submission's workload score, ensuring robustness against stochasticity. Across all workloads, including held-out ones, at least one submission reliably reached the target, demonstrating that none of the workloads were impossible to handle, rather it was difficult for a submission to do well on all of them simultaneously.

**Speed-up comparisons.** In addition to benchmark scores, we compute raw speed-ups between submissions. For relative training speed-ups, we focus on the fixed base workloads and impute infinite workload scores with the ruleset's runtime budget, i.e. assuming the submission would have

achieved the target just after the (artificial) cut-off. This is a best-case assumption for submissions that do not train successfully (and thus a worst-case assumption for speed-ups over those submissions), which may be reasonable in some cases (e.g. Figure 2) but can significantly underestimate actual training times in others. For speed-up calculations, we do not enforce the held-out workload rules that would invalidate workload runtimes when the corresponding held-out workload was not successfully trained. Table 6 lists the average speed-ups over the baseline across all eight base workloads. PYTORCH DISTRIBUTED SHAMPOO provides significantly accelerated training than the external tuning BASELINE, with an average speed-up of $\approx 28\%$. It was also $\approx 2\%$ and $19\%$ faster compared to the second- and third-place submissions, respectively. In the self-tuning ruleset, SCHEDULE FREE ADAMW provides $\approx 8\%$ faster hyperparameter-free training than the self-tuning baseline. While speed-up metrics offer a more intuitive measure, they are less meaningful overall, as the penalty for missing a target is relatively weak. For instance, CASPR ADAPTIVE achieved a finite workload score on only four out of eight workloads, but ranked third in average speed-up.

## 3.2 COMPARISON ACROSS HYPERPARAMETER TUNING RULESETS

To compare submissions across rulesets, we evaluated the first-place self-tuning submission as an external tuning submission. This quantifies the performance gap between rulesets and estimates the expected slowdown from the lack of parallel hyperparameter tuning. The self-tuning winner, the (self-tuning) SCHEDULE FREE ADAMW submission, would have scored a $0.4804$ under the external tuning ruleset, ranking hypothetically eighth, without affecting the scores of other external tuning submissions. Although there is a notable gap compared to its external tuning counterpart, it is surprisingly competitive in this ruleset, given its lack of parallel tuning. Across the seven base workloads both submissions trained successfully, PYTORCH DISTRIBUTED SHAMPOO, the winner of the external tuning ruleset, is $\approx 24\%$ faster than (self-tuning) SCHEDULE FREE ADAMW. However, (self-tuning) SCHEDULE FREE ADAMW is $\approx 10\%$ faster than the (external tuning) BASELINE across the seven base workloads both trained successfully. In other words, (self-tuning) SCHEDULE FREE ADAMW reduced training time by $10\%$ compared to the baseline, using only a single machine instead of five in parallel exploring different hyperparameter configurations. Although there is still a meaningful gap between the rulesets, the encouraging results of (self-tuning) SCHEDULE FREE ADAMW suggest that even without any free workload-specific tuning, it might be possible to create training algorithms that are surprisingly effective.

## 3.3 RULE CHANGE COUNTERFACTUALS

In order to better understand the effects of the specific ALGOPERF benchmark rules, we can consider how the results would have changed with different rules.

**Different ways of determining whether a submission receives a finite runtime score.** There are a variety of mechanisms in the rules whereby a submission can fail to receive a finite runtime score on a given workload, and they triggered relatively frequently in the competition. In the external tuning ruleset, out of 154 workload-submission combinations (including both fixed base and held-out workloads), 33 ($\approx 21\%$)were classified as `infs` due to a submission not reaching the target (31 out of 84 or $\approx 37\%$ for the self-tuning ruleset), while 5 ($\approx 3\%$) and 4 ($\approx 5\%$), respectively, were classified as `NaNs` (error before first evaluation). Additionally, 8 held-out workload scores ($\approx 12\%$) in the external tuning ruleset were ignored due to failure to reach the target on the base workload, while the reverse occurred for 7 base workload scores ($\approx 8\%$). In the self-tuning ruleset, 3 held-out scores ($\approx 8\%$) and 2 base scores ($\approx 4\%$) were excluded for the same reasons. Finally, 11 scores ($\approx 7\%$) in the external and 4 scores ($\approx 5\%$) in the self-tuning ruleset exceeded the four-times-the-fastest-valid-runtime threshold and were thus classified as infinite. This is most notable for OGBG, where the fast training times of CASPR ADAPTIVE invalidated a significant number of workload scores. Ultimately, requiring training algorithms to be robust to workload variations through the held-out workloads *did* provide a non-trivial constraint, though the held-out workload targets were not so stringent the effect was that large. Only considering submissions to train a workload successfully when no other submission could train more than four times faster, primarily affected workloads where the overall runtime budget was the most generous (in hindsight). This rule might become redundant if runtime budgets for future iterations are tightened based on the top-performing submissions. In Figure 3 (Appendix A.5), we investigate the sensitivity of benchmark scores and rankings to changes in $\tau_{\max}$, the upper limit of the performance profile.

**Scoring on a subset of workloads.** How would the competition results change if we considered only a subset of the benchmark's workloads? Figure 4 shows the performance profiles and benchmark scores when ignoring all held-out workloads. With the exception of CASPR ADAPTIVE and AMOS, which switch positions, held-out workloads have little impact on the leaderboard. However, quantifying the full effect of held-out workloads is challenging. Their presence may have discouraged submissions overfitting to the fixed workloads. While held-out workloads offer some value, their costs likely outweigh their benefits (Section 5). These costs include the additional compute and, perhaps more importantly, the significant human effort involved in designing and implementing them.

Dahl et al. (2023) envisioned ALGOPERF's qualification set as a cheaper way to evaluate training algorithms, allowing for better allocation of compute resources to the more promising submissions. For the competition, we had enough compute to fully score all submitted training algorithms and did not use the qualification set (for a description of all changes between the benchmark and competition see Appendix A.2). Figure 5 shows the submissions' score when using only the qualification set (which also excludes held-out workloads). While the qualification set helps to identify under-performing submissions, rankings can change substantially compared to the full set. We can also assess how scores shift when removing individual workloads. Table 7 presents the benchmark scores and rankings when each workload is excluded. The rankings remain largely stable, with the winning submissions unchanged in all but one case, indicating that the competition results are robust to the precise selection of workloads, as long as there is a large and diverse enough set.

## 4 ENGINEERING CHALLENGES

Scoring submissions based on wall-clock time, especially in a multi-framework setting, introduces non-trivial engineering challenges. Requiring competing submissions to make use of shared workload implementations and timing code is an essential requirement for meaningful timing measurements, but also introduces a host of issues. Even for standard algorithms, seemingly minor implementation details can have a dramatic effect on time-per-step or how quickly the loss decreases per step. As much as possible, potential confounding factors for training algorithm comparisons should be removed, while still allowing realistic participation. Perhaps the biggest point of tension between controlling measurements and remaining relevant to current practice is due to supporting multiple frameworks. Submissions in JAX & PYTORCH have to be compared on a level playing field, despite framework-specific differences in optimization and execution paradigms. To ensure as fair a comparison as possible between algorithms implemented in JAX & PYTORCH, the workload implementations in each framework should be functionally equivalent, high-quality, and realistically performant:

**Functional equivalence**: workload implementations should perform mathematically identical computations across frameworks and use identical implementation strategies. Given the same input (batch, model parameters, random initialization), the outputs of the forward and backward passes should be the same (within numerical precision) regardless of whether the workload is implemented in JAX or PYTORCH. All parts of the computation not controlled by the submission code (everything from forward & backward passes to weight initialization to timing & logging) should be semantically identical and also be parallelized in the same way on the same devices. This guarantees that any observed performance differences are truly due to algorithmic improvements within the submissions, and not variations in the underlying computations.

**Performant in both time and memory**: workload implementations shouldn't saddle training algorithm submissions with time or memory overheads that skilled engineers would easily avoid outside the context of the competition. Although achieving identical execution times (and memory usage patterns) across frameworks when run on identical competition hardware is not required in principle (and is nearly impossible), in practice since both JAX & PYTORCH are mature frameworks and the competition workloads are standard deep learning benchmark problems, any large discrepancy between time or space efficiency across workload implementations in different frameworks is cause for concern. Hypothetically, different frameworks can make different techniques easier or harder to express or make achieving certain memory or runtime requirements more or less difficult. However, only when the workload implementations in *both* frameworks are as efficient as is realistically possible can one begin to ask how the constraints of the frameworks themselves are playing a role. Implementations should realistically capture how the frameworks are used by practitioners while also being as efficient as possible.

Addressing these challenges led to the identification and subsequent implementation of various improvements and best practices. Our experience should be useful for researchers attempting to make similar reproducible measurements with as few confounding factors as possible.

## 4.1 FUNCTIONAL EQUIVALENCE OF JAX & PYTORCH IMPLEMENTATIONS

**Framework-specific defaults.** JAX & PYTORCH provide basic primitives for implementing various components in neural networks. Creating functionally equivalent workloads requires considerable care around framework specific features and defaults. For example, JAX's default `Gelu` activation function calculates the cumulative density function of the Gaussian with a `Tanh` approximation, whereas in PYTORCH the exact computation is used. Similarly, for layer normalization, the default values for $\varepsilon$ differ between the frameworks. Even more insidious, the weights of linear layers in JAX are initialized with `lecun_normal`, while similar weights in PYTORCH are initialized with `kaiming_uniform`. Moreover, PYTORCH does not supply a `lecun_normal` initialization, necessitating a manual implementation.

**Data pre-processing and augmentation.** Differences in data pre-processing and augmentation strategies can result in data pipelines that are not comparable. To mitigate these types of differences, the same TENSORFLOW (Abadi et al., 2015) dataset pipelines were used across frameworks where possible (CRITEO 1TB, FASTMRI, LIBRISPEECH, OGBG, and WMT). However, the IMAGENET VIT and RESNET workloads in PYTORCH use a custom implementation to match the TENSORFLOW random augmentation strategy in the JAX workload.

## 4.2 PERFORMANCE OF JAX & PYTORCH IMPLEMENTATIONS

**Data pipelines.** Differences in the data parallelism implementations may also lead to additional overhead for PYTORCH workloads. To match JAX's data parallel paradigm, the PYTORCH workloads are implemented with `DistributedDataParallel` (DDP). DDP requires one PYTHON process per device, which in a naive implementation would lead to replication of TENSORFLOW data pipeline tasks in each process and potentially use too much RAM and an undesirable number of threads. To mitigate this issue, TENSORFLOW data pipeline operations are only run in one PYTHON process and batches are broadcasted to the remaining processes. This choice results in a small additional communication overhead for each batch in the PYTORCH workloads.

**Custom CUDA kernels.** The wall-clock time of two functionally equivalent computations can significantly differ due to differences in the underlying GPU implementations. One example for this is the `torch.lstm` implementation. In PYTORCH, the LSTM implementation uses a custom CUDA kernel that results in a $2\times$ smaller wall-clock time on the LIBRISPEECH DEEPSPEECH workload compared to the JAX version. This uncovered the opportunity to generate a similar CUDA kernel in the JAX LSTM layer implementation. Using the updated faster JAX LSTM layer implementation significantly sped up the end-to-end wall clock times for the DEEPSPEECH workload. [6]

**Adopting PYTORCH 2.0 features** Achieving realistically performant workload implementations requires adopting novel features and best practices in both frameworks. Without these, the performance gaps between PYTORCH & JAX could be as large as 60%. Over the course of the development of the competition, the performance of PYTORCH workload implementations significantly improved as a result of migrating from PYTORCH to PYTORCH 2.0 (see Appendix A.6).

PYTORCH 2.0 (Ansel et al., 2024), released in Dec 2022, introduced two major extensions that represent a major departure from PYTORCH's (Paszke et al., 2019) original eager programming model, where every line of code would dispatch to a CUDA kernel. Specifically, PYTORCH 2.0 introduced `TorchDynamo`, a Python level JIT compiler that enables graph compilation of PYTORCH programs, and `TorchInductor`, a compiler backend which translates PYTORCH programs into Triton (Tillet et al., 2019) kernels for GPU and C++ for CPUs. The main adjustments that had to be made to adopt to PYTORCH 2.0 and close the speed gap to JAX were related to `torch.compile`:

- **Avoiding graph breaks.** When `TorchDynamo` encounters unsupported functionality, it creates a graph break, splitting the model. To maximize performance, graph breaks should be prevented and models compiled with `torch.compile(model, fullgraph=True)`.

---

[6]https://github.com/google/jax/pull/13319

- **Compiling the loss function, not just the model.** Most of the tutorials surrounding `torch.compile` implied that it is only to be applied to the model by running `torch.compile(model)`. However, compiling loss functions had a dramatic impact on performance for several workloads. [7]
- **Overhead reduction mode.** For models where overhead reduction is crucial it is recommended to use CUDA graphs. This can be enabled via `torch.compile(model, mode="reduce-overhead")` for the cost of a small memory overhead.
- **Combining `DDP` and `torch.compile`**: The compiler also needs to compose with the other subsystems, e.g., there are subtle differences between `ddp(torch.compile(model))` and `torch.compile(ddp(model))`. With the latter, the compiler will also trace collectives which may result in further performance optimizations.

**Memory allocator settings** Older CUDA versions are more likely to cause OOM errors in PY-TORCH. Setting `torch.cuda.memory._set_allocator_settings('expandable_-segments:True')` can fix OOMs caused by memory fragmentation. [8]

Making adjustments to use modern features and adopting best practices for PYTORCH & JAX helped achieve realistically performant workload implementations in both frameworks and significantly reduced the time gap in between the JAX & PYTORCH workload implementations. While performance differences still exist, they are mostly within tolerance (12%) and are balanced such that neither framework has an overwhelming advantage. Measurements of the performances gaps between PYTORCH & JAX workloads and improvements can be found in Appendix A.6.

## 5 LESSONS LEARNED

The improved training algorithms developed for the competition delivered significant speedups in neural network training. The winning submissions achieved 28% and 8% faster model training compared to their respective baselines. The winning submission in the external tuning ruleset, based on DISTRIBUTED SHAMPOO, demonstrates that non-diagonal preconditioning methods can improve runtime over currently popular methods like ADAM. And yet, **despite these significant strides made in accelerating neural network training, there is ample room left for algorithmic improvement**. No single submission dominated across all workloads; instead, five different submissions achieved the best performance on at least one of the eight base workloads.

**These algorithmic advances can only be realized reliably by careful benchmarking and engineering efforts ensuring fair and meaningful comparisons**. Comparisons on multiple deep learning workloads are required to isolate a robust signal of a submission's performance. Seemingly intuitive aggregate metrics, like average speedup, fail to fully capture pertinent aspects of a training algorithm's practical usefulness. Precise engineering work is required to ensure that training algorithms are compared fairly. In Section 4, we identified implementation details that dramatically affect the runtime across virtually all training algorithms, and numerous potentially confounding factors that must be accounted for in algorithmic comparisons, in particular across deep learning frameworks.

The competition also underscores the inherent link between hyperparameter tuning and training algorithm performance. **To meaningfully compare training algorithms, measurements must properly account for workload-specific hyperparameter tuning and training algorithms must be fully-specified without leaving free parameters for the user to set.** A complete training algorithm specification includes a formal specification of a training algorithm's hyperparameter defaults and/or search space, and should include regularization choices. The varying performance of many submissions across workloads suggests that hyperparameter tuning remains a significant challenge; the top-scoring submissions mostly distinguished themselves by their reliable training across a large variety of workloads. Despite promising advances in hyperparameter-free algorithms (as seen in the self-tuning ruleset), fully-automatic neural network training remains a serious challenge. In light of our results, future publications of training algorithms should include clear hyperparameter (search space) recommendations and be paired with a recommended tuning protocol, ideally one that is sensitive to a user-supplied tuning budget. Although a radical change from the current practice of published training algorithms that aren't runnable without setting various hyperparameters, publishing

---

[7]https://github.com/mlcommons/algorithmic-efficiency/pull/597
[8]https://github.com/mlcommons/algorithmic-efficiency/issues/497

families of update rules and abdicating responsibility for tuning entirely to the user only adds to the community's confusion on what to actually use.

## 5.1 METHODOLOGICAL LESSONS

In our competition, the ALGOPERF: TRAINING ALGORITHMS benchmark has proven to be quite effective in differentiating algorithms and measuring progress in neural network training, but it results in quite an involved experimental protocol with substantial costs. Most of its features, e.g. (integrated) performance profiles, were useful in order to generate nuanced and robust insights. However, based on our experience, we propose the following modifications to the ALGOPERF benchmark going forward: (1) **Removing held-out workloads.** Eliminating the held-out workloads would drastically simplify the evaluation process and reduce the benchmark's runtime substantially. Though held-out workloads have the potential to deter overfitting to the base workloads, they require substantial computational, logistical (they can't be generated until submissions are frozen), and engineering effort. Replacing the six held-out workloads with one or two additional base workloads would provide similar benefits, while reducing runtime and allowing additional, practically-relevant workloads to be included, such as autoregressive language models or diffusion models. (2) **Reducing the runtime budgets to reduce costs, especially for the self-tuning ruleset.** The submissions have demonstrated that, for many workloads, significantly less training time is needed. Matching the self-tuning budget to the current external tuning budget, and potentially reducing the external tuning budget further, would maintain meaningful comparisons while lowering overall costs. (3) **Reducing the number of studies from 5 to 3** would cuts compute costs by an additional 40%. These repetitions with different random seeds ensure robust insights rather than random noise due to the stochastic training process. However, our results indicate fewer studies are sufficient for statistical fidelity. Additionally, a modernized hardware setup with a better cost-to-performance ratio could further reduce costs.

## 6 CONCLUSION

An inescapable limitation of empirical comparisons of (training) algorithms (see also Choi et al., 2019; Sivaprasad et al., 2020; Schmidt et al., 2021) is that one can only directly measure the behavior of specific implementations, not the abstract algorithmic ideas they express. The ALGOPERF: TRAINING ALGORITHMS competition embraces this constraint by evaluating submissions that fully specify everything necessary to apply them to any generic learning problem, including all necessary workload-specific hyperparameter tuning. The competition conditions provide a realistic simulation of applying a generic training algorithm to a new problem, with some important caveats. First and foremost, the maximum runtime budgets and evaluation metric targets for each workload provide a lot of information about what training horizon is appropriate and what validation error is achievable, unlike in a novel learning task where such information isn't available. Second, by scoring on time to reach particular *validation* error goals and not using results on separate per-workload test sets, the competition conditions do not capture the challenges of train/test skew or distribution shift. Additionally, while this competition reveals *how well* training algorithms perform, deeper analysis, such as Kunstner et al. (2024), is needed to explain *why*.

Nevertheless, despite its limitations, the ALGOPERF: TRAINING ALGORITHMS competition has produced insights about the current training algorithm landscape. In particular, the competition results have validated two exciting directions for improvements: non-diagonal preconditioning (PYTORCH DISTRIBUTED SHAMPOO) and hyperparameter reduction strategies (SCHEDULE FREE ADAMW). These insights are only possible because they are the result of an open and *competitive* process. This is in contrast to existing works that, due to the lack of a suitable existing benchmark, have each been forced to introduce their own training algorithm evaluation protocols. As a result, much of existing research on training algorithms has focused on isolating and improving individual ideas and components, but hasn't always produced complete and usable training algorithms that incorporate viable hyperparameter tuning protocols, as would be necessary to perform well in competition.

Our end goal is not the specific insights generated by the inaugural round of comparisons, as interesting as they might be. Ideally, competitions like ALGOPERF will affect training algorithms research in two main ways. First, they should provide a sieve to filter the most useful ideas out of the literature by separating truly practical methods from interesting ideas that aren't yet useful. Second, they should provide a valuable signal to guide the design process for new training algorithms.

ACKNOWLEDGMENTS

FS was supported by funds from the Cyber Valley Research Fund. PH and FS gratefully acknowledge co-funding by the European Union (ERC, ANUBIS, 101123955), and by the DFG through Project HE 7114/5-1 in SPP2298/1; PH is a member of the Machine Learning Cluster of Excellence, funded by the DFG under Germany's Excellence Strategy – EXC number 2064/1 – Project number 390727645; PH and FS also acknowledge the German Federal Ministry of Education and Research (BMBF) through the Tübingen AI Center (FKZ:01IS18039A); and funds from the Ministry of Science, Research and Arts of the State of Baden-Württemberg.

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

# A APPENDIX

## A.1 ALGOPERF DETAILS

In this section, we provide additional details about the ALGOPERF: TRAINING ALGORITHMS benchmark (Dahl et al., 2023) and our competition based on it. Tables 2 and 3 summarize the fixed base and held-out workloads of the ALGOPERF benchmark as they are used in the competition.

**Table 2: Summary of *fixed* base workloads in the ALGOPERF benchmark.** Losses include cross-entropy (CE), mean absolute error (L1), and Connectionist Temporal Classification loss (CTC). Additional evaluation metrics are structural similarity index measure (SSIM), (word) error rate (ER & WER), mean average precision (mAP), and bilingual evaluation understudy score (BLEU). Note: Some workloads have minor changes (see Appendix A.2) to the runtime budgets and validation targets compared to the ALGOPERF benchmark publication (Dahl et al., 2023). The runtime budget is that of the external tuning ruleset, the self-tuning ruleset allows 3× longer training.

| Task | Dataset | Model | Loss | Metric | Validation Target | Runtime Budget |
|---|---|---|---|---|---|---|
| Clickthrough rate prediction | CRITEO 1TB | DLRMSMALL | CE | CE | 0.123735 | 7703 |
| MRI reconstruction | FASTMRI | U-NET | L1 | SSIM | 0.7344 | 8859 |
| Image classification | IMAGENET | RESNET-50 | CE | ER | 0.22569 | 63,008 |
| | | VIT | CE | ER | 0.22691 | 77,520 |
| Speech recognition | LIBRISPEECH | CONFORMER | CTC | WER | 0.085884 | 61,068 |
| | | DEEPSPEECH | CTC | WER | 0.119936 | 55,506 |
| Molecular property prediction | OGBG | GNN | CE | mAP | 0.28098 | 18,477 |
| Translation | WMT | TRANSFORMER | CE | BLEU | 30.8491 | 48,151 |

**Table 3: Summary of *held-out* workloads sampled for this iteration of the ALGOPERF competition.** One held-out workload was sampled from the random workload variants (see (Dahl et al., 2023, Table 11)) for each dataset used in the fixed workloads. Note, for the IMAGENET and LIBRISPEECH datasets only a single held-out workload was sampled, according to the modified competition rules (see Appendix A.2). The loss function and performance metrics are identical to the corresponding fixed base workload (Table 2).

| Task | Dataset | Variant | Validation Target | Runtime Budget |
|---|---|---|---|---|
| Clickthrough rate prediction | CRITEO 1TB | EMBED INIT SCALE | 0.129657 | 7703 |
| MRI reconstruction | FASTMRI | TANH | 0.717840 | 8859 |
| Image classification | IMAGENET | RESNET BN INIT SCALE | 0.23474 | 63,008 |
| Speech recognition | LIBRISPEECH | CONFORMER LAYERNORM | 0.09731 | 61,068 |
| Molecular property prediction | OGBG | ALTERED LAYERS | 0.269446 | 18,477 |
| Translation | WMT | GLU & TANH | 29.5779 | 48,151 |

## A.2 MODIFICATIONS TO THE ALGOPERF BENCHMARK FOR THIS COMPETITION

For the purposes of this competition, we made several modifications to the benchmark from how it was initially proposed by Dahl et al. (2023), mainly to reduce the competition's overall compute costs. All modifications were done before the competition's submission deadline. Most notably, the number of tuning trials in the external tuning ruleset was reduced from 20 to 5. This change significantly lowers the computational cost of the competition while also mitigating the risk of submissions overfitting to the benchmark's workload selection. To further reduce computational

costs, we also decreased the runtime budgets for both LIBRISPEECH workloads significantly—from $101,780$ to $61,068$ seconds and from $92,509$ to $55,506$ seconds—with only a slight impact on target performance (the WER changed from $0.078477$ to $0.085884$ and from $0.1162$ to $0.119936$). Additionally, small bug fixes led to slight changes in two other workload targets: the target loss on CRITEO 1TB increased from $0.123649$ to $0.123735$, and the target SSIM on FASTMRI decreased from $0.7344$ to $0.723653$. Furthermore, only a single held-out workload was sampled for each *dataset*, not each workload. This means that for both the IMAGENET and LIBRISPEECH datasets, only one, instead of two, held-out workloads were sampled from the combined set of workload variants, further reducing the competition's overall runtime significantly.

Finally, we decided to use only the validation targets, excluding the test set targets. Including both validation and test set targets, as proposed by Dahl et al. (2023), complicates the evaluation process by involving the practitioner's task of selecting hyperparameters for an unknown test set based on validation performance. We chose to omit this aspect of the training process in the current iteration of the benchmark.

### A.3 COMPUTATIONAL COSTS

The ALGOPERF benchmark requires a substantial number of training runs and thus significant computational resources to yield meaningful results. To evaluate a single submission, it needs to be evaluated on eight fixed—and six held-out—workloads. For each workload, five repetition studies are performed, and in the external tuning ruleset, each study comprises five tuning trials. In this iteration of the competition, we conducted 3850 runs in the external tuning ruleset (due to eleven training algorithms evaluated on 14 workloads, with $5 \cdot 5$ trials) and 420 runs in the self-tuning ruleset (six training algorithms evaluated five times on 14 workloads, with a runtime budget $3 \times$ larger). The actual computational cost of these 4000+ training runs varies significantly based on the performance of the submission. Training can terminate early if the validation (and test targets)[9] are met or if errors occur, such as out-of-memory (OOM) issues. The runtime budget in Table 2 reflects only the compute time spent by the submissions themselves, excluding time for free evaluations or other operations outside the submission functions. It is thus not an upper bound on the total computational cost. On average, we required approximately $1847$ h to fully run a self-tuning submission and $3469$ h per external tuning submission, amounting to a total of roughly $49,240$ h on the competition hardware ($8 \times$ NVIDIA V100 GPUs). This is considerably less than what is reported in Dahl et al. (2023, Table 15), primarily due to fewer tuning trials and a reduced runtime budget for certain workloads (see Appendix A.2), as well as early termination of some submissions. In theory, the number of required runs could be reduced, for example, by skipping a held-out workload if a submission fails the corresponding fixed workload. However, this would limit the ability to parallelize different runs and reduce opportunities for further analysis beyond calculating benchmark scores.

### A.4 SUBMISSION DETAILS

In this section, we provide details on the submissions received in this iteration of the competition. Table 4 lists the submissions to the external tuning ruleset, while Table 5 details submissions received for the self-tuning ruleset. Notably, no submission in this iteration significantly modified the `data_-selection` function, indicating a potential area for future exploration.

### A.5 ADDITIONAL COMPETITION RESULTS

In this section, we provide additional analysis of the ALGOPERF competition results. In Figure 2, we investigate submissions could achieve the target performance on the RESNET workload at least once, but not reliable enough to receive a finite score. Table 6 compares the training time speed-ups of all submission relative to the BASELINE in their respective ruleset. We investigated the sensitivity of leaderboard rankings and benchmark scores to changes in $\tau_{\max}$, the upper limit of the performance profile and integration for the benchmark score. As shown in Figure 3, rankings remain relatively stable for most submissions across different $\tau_{\max}$ values. With Figures 4 and 5 and Table 7 we explore how hypothetical rules changes would affect the competition results.

---

[9]Although we decided not to use test targets for the purpose of our competition, we wanted to log when submissions reached the test targets for potential future analysis.

**Table 4: External tuning submissions.** Details of all submissions to the external tuning ruleset.

| Submission | Authors | Institutions | Framework | Description |
|---|---|---|---|---|
| PYTORCH DISTR. SUBMISSION | Hao-Jun Shi, Tsung-Hsien Lee, Anna Cai, Shintaro Iwasaki, Wenyin Fu, Yuchen Hao, Mike Rabbat | Meta Platforms | PYTORCH | Based on the Distributed Shampoo algorithm of Anil et al. (2020) with an implementation tailored to leverage PyTorch performance optimizations. See Shi et al. (2023) for details. The submission uses a list of five hyperparameter settings. |
| SCHEDULE FREE ADAMW | Alice Yang, Aaron Defazio, Konstantin Mishchenko | Meta AI, Samsung AI | PYTORCH | A externally tuned version of SCHEDULE FREE ADAMW (Defazio et al., 2024) with a list of five hyperparameter configurations. |
| GENERALIZED ADAM | George Dahl, Sourabh Medapati, Zack Nado, Rohan Anil, Shankar Krishnan, Naman Agarwal, Priya Kasimbeg, Vlad Feinberg | Google | JAX | Submission with an ADAM-style update rule, tuning over the use of Nesterov acceleration and preconditioning. Essentially tuning over ADAMW (Kingma & Ba, 2015), NADAMW, and SGD (Robbins & Monro, 1951) with or without momentum. |
| CYCLIC LR | Niccolò Ajroldi, Antonio Orvieto, Jonas Geiping | MPI-IS, ELLIS Institute Tübingen | PYTORCH | Revisits the work of Loshchilov & Hutter (2017) and Smith (2017), coupling NADAMW (Dozat, 2016; Loshchilov & Hutter, 2019) with a cyclic learning rate scheduler. Each cycle involves a linear warmup phase for the LR, followed by cosine annealing. |
| NADAMP | George Dahl, Sourabh Medapati, Zack Nado, Rohan Anil, Shankar Krishnan, Naman Agarwal, Priya Kasimbeg, Vlad Feinberg | Google | JAX | Uses NADAMW with an extra tunable hyperparameter $p$ enabling $p$th root of denominator inside NADAMW update rule instead of the default of 2. |
| BASELINE | | | JAX | Baseline using NADAMW (Dozat, 2016; Loshchilov & Hutter, 2019) and a linear learning rate warmup followed by a cosine decay (Dahl et al., 2023). |
| AMOS | Ran Tian | Google | JAX | Submission based on the AMOS optimizer (Tian & Parikh, 2022) with a list of five hyperparameter settings. |
| CASPR ADAPTIVE | Sai Surya Duvvuri, Inderjit S. Dhillon, Cho-Jui Hsieh | UT Austin, UCLA, Google | JAX | A submission based on (Duvvuri et al., 2024) with a list of five hyperparameter configurations. |
| LAWA QUEUE | Niccolò Ajroldi, Antonio Orvieto, Jonas Geiping | MPI-IS, ELLIS Institute Tübingen | PYTORCH | Employs Latest Weight Averaging (Izmailov et al., 2018; Kaddour, 2022) on top of NADAMW (Dozat, 2016; Loshchilov & Hutter, 2019), maintaining a queue of previous model weights. The queue is periodically updated during training and passed to the competition API for evaluation. |
| LAWA EMA | Niccolò Ajroldi, Antonio Orvieto, Jonas Geiping | MPI-IS, ELLIS Institute Tübingen | PYTORCH | Similar to LAWA QUEUE but maintaining an exponential moving average of the model weights, which is updated periodically during training and returned to the competition API for evaluation. |
| SCHEDULE FREE PRODIGY | Alice Yang, Aaron Defazio, Konstantin Mishchenko | Meta AI, Samsung AI | PYTORCH | Combining Schedule-free (Defazio et al., 2024) with the PRODIGY optimizer (Mishchenko & Defazio, 2024). |

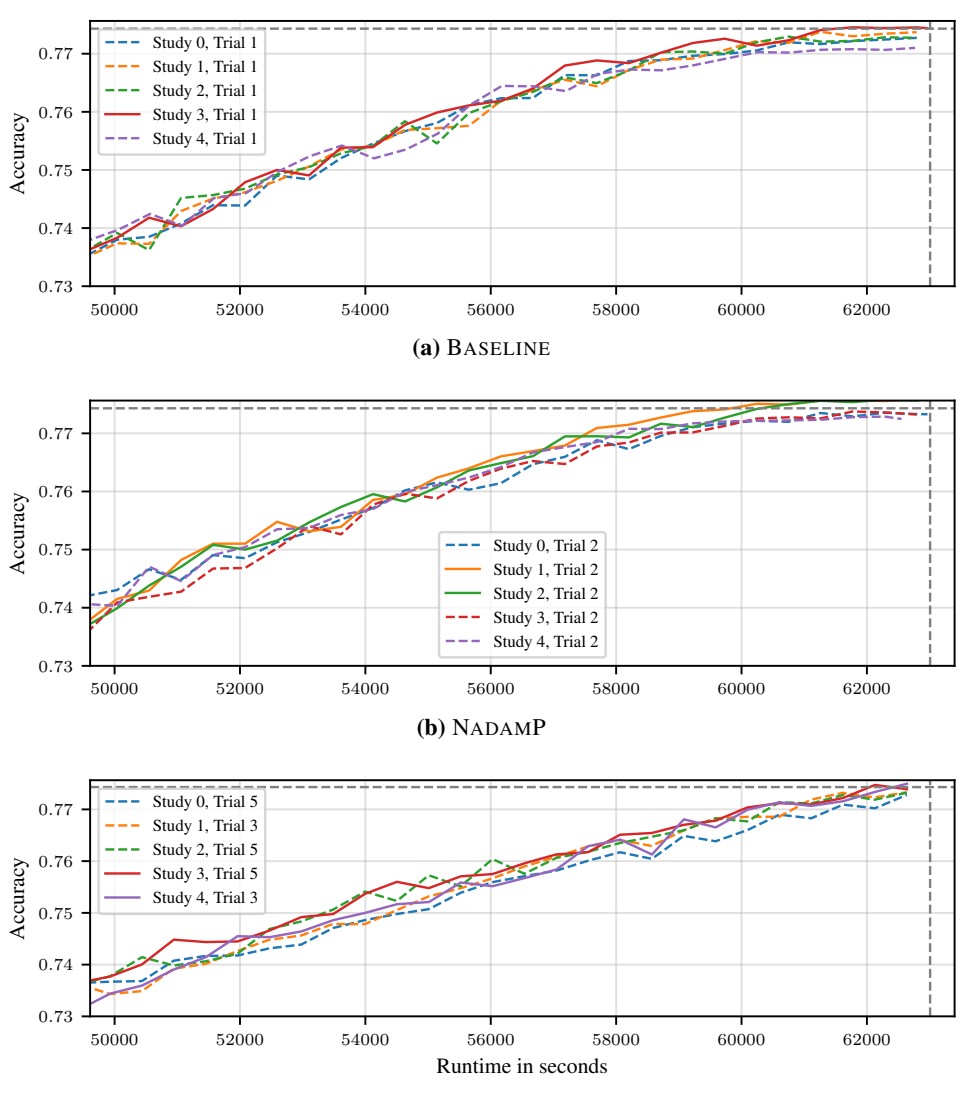

**(a)** BASELINE

**(b)** NADAMP

**(c)** PYTORCH DISTRIBUTED SHAMPOO

**Figure 2: Validation accuracy vs. runtime on the RESNET workload.** The BASELINE (Figure 2a), NADAMP (Figure 2b), and PYTORCH DISTRIBUTED SHAMPOO (Figure 2c) all reach the validation target on the RESNET workload for at least one study but not reliably enough to get a finite score. Shown are the best trials from each of the five studies, where "best" is either the fastest trial to achieve the target performance or the trial whose best performance is closest to the target. Trials that reach the target are marked with a solid line, while studies that do not reach the target are indicated with a dashed line. The gray dashed horizontal and vertical lines indicate the target performance and runtime budget respectively. Additionally, both AMOS and CYCLIC LR came close but missed the target in all studies.

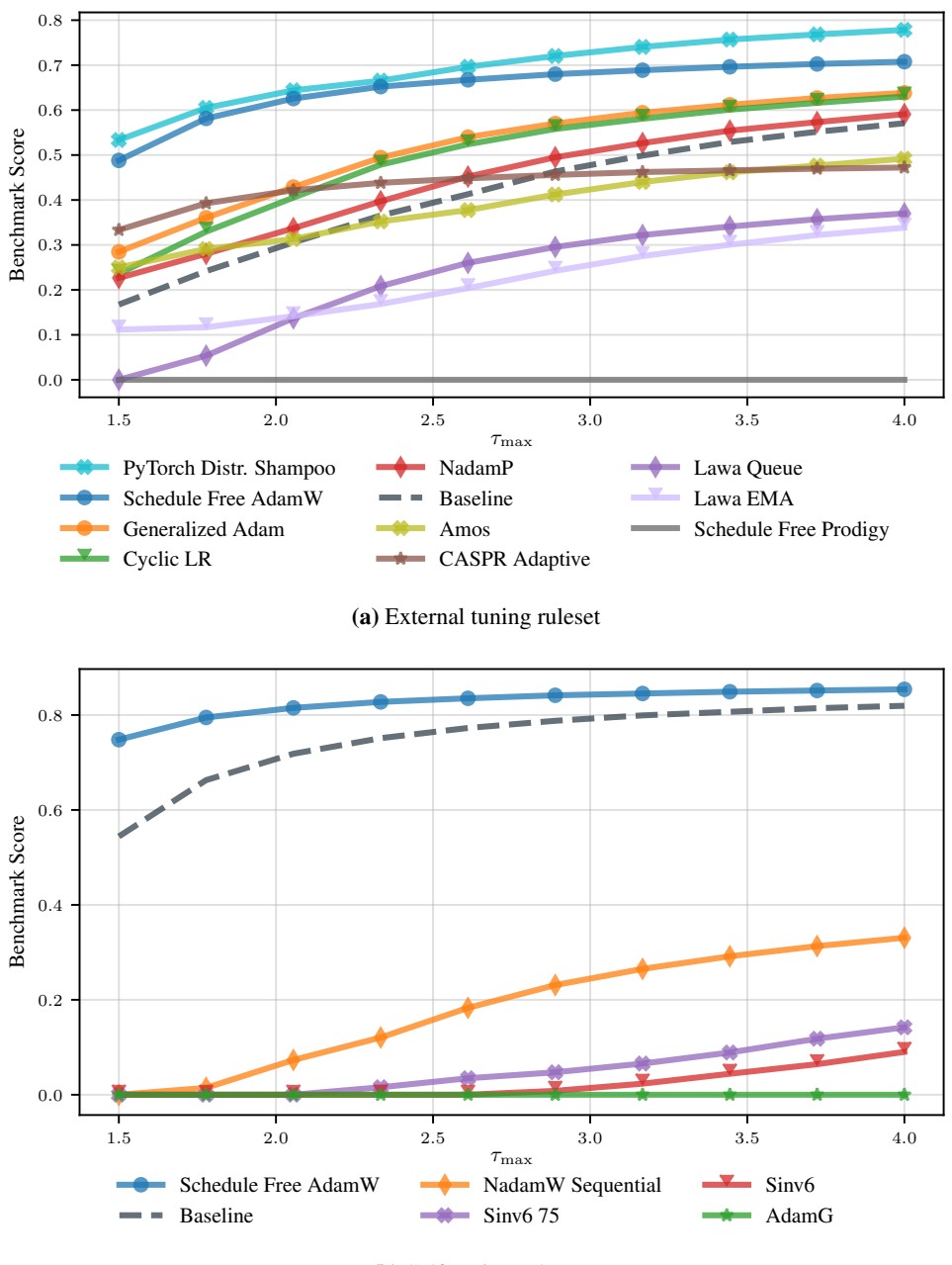

**(a)** External tuning ruleset

**(b)** Self-tuning ruleset

**Figure 3: Benchmark score as a function of $\tau_{\max}$.** The upper limit of the performance profile and upper integration limit for the benchmark score, $\tau_{\max}$, determines which workload scores are treated as finite and influences the penalty for infinite scores. We observe that rankings remain stable for most submissions across different values of $\tau_{\max}$.

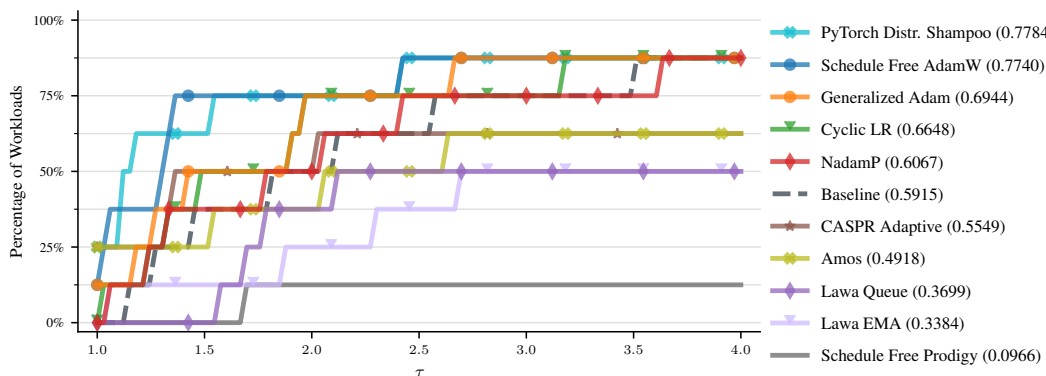

**(a)** Performance profiles for the external tuning ruleset ignoring held-out workloads

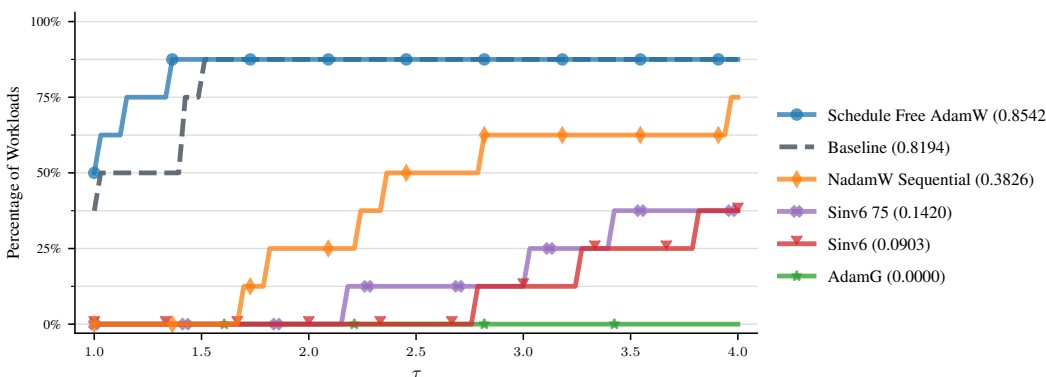

**(b)** Performance profiles for the self-tuning ruleset ignoring held-out workloads

**Figure 4: Performance profiles of all ALGOPERF submissions when ignoring held-out workloads.** Structurally the same as Figure 1 but here we ignore all benchmark rules involving the held-out workloads.

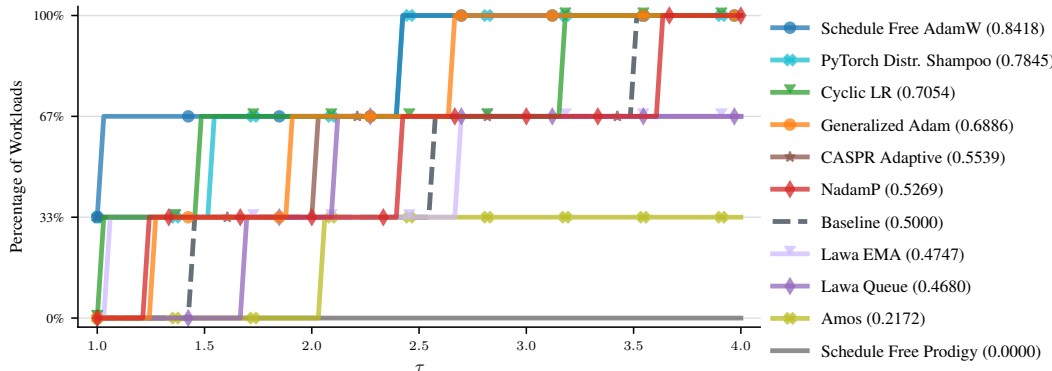

**(a)** Performance profiles for the external tuning ruleset on the qualification workloads

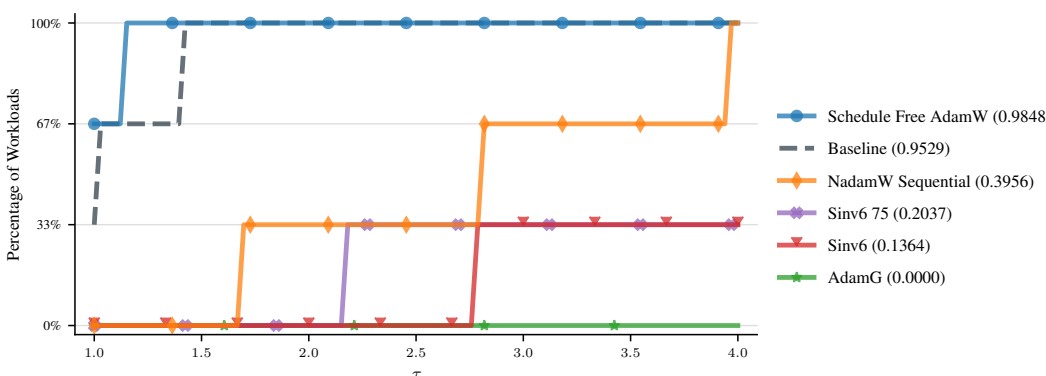

**(b)** Performance profiles for the self-tuning ruleset on the qualification workloads

**Figure 5: Performance profiles of all ALGOPERF submissions on the qualification workloads.** Structurally the same as Figure 1 and Figure 4 but only considering the three workloads that are part of the qualification set, i.e. CRITEO 1TB, WMT, and OGBG. In the qualification set, no held-out workloads are used.

**Table 5: Self-tuning submissions.** Details of all submissions to the external tuning ruleset.

| Submission | Authors | Institutions | Framework | Description |
|---|---|---|---|---|
| SCHEDULE FREE ADAMW | Alice Yang, Aaron Defazio, Konstantin Mishchenko | Meta AI, Samsung AI | PYTORCH | A self-tuning version of SCHEDULE FREE ADAMW (Defazio et al., 2024) using a single hyperparameter configuration. |
| BASELINE | | | JAX | Baseline using NADAMW, a linear learning rate warmup followed by a cosine decay, and a single hyperparameter point (Dahl et al., 2023). |
| NADAMW SEQUENTIAL | George Dahl, Sourabh Medapati, Zack Nado, Rohan Anil, Shankar Krishnan, Naman Agarwal, Priya Kasimbeg, Vlad Feinberg | Google | JAX | Uses NADAMW update rule and runs 3 fixed hyperparameter points sequentially. The intention was for these to be the top 3 hyperparameter points found at one third the self-tuning ruleset step budgets. |
| SINV6 75 | Abhinav Moudgil | Mila, Concordia University | JAX | A submission for a task-invariant learned optimizer meta-trained on small tasks. Uses 75% of the number of steps as target in learned optimizer initialization. |
| SINV6 | Abhinav Moudgil | Mila, Concordia University | JAX | A submission for a task-invariant learned optimizer meta-trained on small tasks. |
| ADAMG | Yijiang Pang | Michigan State University | PYTORCH | A submission based on the ADAMG optimizer (Pang et al., 2024). |

**Table 6: Speed-ups vs. the baseline.** To compute the speed-up over the baseline, we first compute the geometric mean of the workload runtimes relative to the runtimes of the baseline. Only fixed base workloads are considered. If a submission did not reach the target on a workload, its runtime is imputed with the runtime budget, i.e. assuming that the submission would have reached the target just after the cut-off. This is the best-case assumption for the submissions. We do not set runtimes to infinity, e.g. because the corresponding held-out workload was not trained successfully. The geometric mean is then expressed as a relative speed-up over the baseline, with positive numbers representing faster training.

**(a) External tuning**

| Submission | Speed-up |
|---|---|
| PYTORCH DISTR. SHAMPOO | 27.87% |
| SCHEDULE FREE ADAMW | 26.60% |
| CASPR ADAPTIVE | 24.33% |
| GENERALIZED ADAM | 10.89% |
| CYCLIC LR | 10.67% |
| LAWA QUEUE | 7.98% |
| NADAMP | 3.37% |
| AMOS | 1.46% |
| BASELINE | 0.00% |
| LAWA EMA | −9.17% |
| SCHEDULE FREE PRODIGY | −15.68% |

**(b) Self-tuning**

| Submission | Speed-up |
|---|---|
| SCHEDULE FREE ADAMW | 7.76% |
| BASELINE | 0.00% |
| NADAMW SEQUENTIAL | −92.44% |
| SINV6 75 | −157.67% |
| SINV6 | −168.63% |
| ADAMG | −294.16% |

**Table 7: Submission benchmark scores and ranking when removing individual workloads.** Shown are the benchmark scores (S.) and leaderboard rankings (R.) of all external tuning (Table 7a) and self-tuning (Table 7b) submissions, when dropping specific workloads. For example, the last column reports the rankings of the submissions if we consider all workloads (including held-out workloads) *except* the WMT workloads. The "Full" columns show the scores and ranks when considering all workloads.

**(a) External tuning ruleset**

| | Full | | CRITEO 1TB | | FASTMRI | | RESNET | | VIT | | CON-FORMER | | DEEP SPEECH | | OGBG | | WMT | |
|---|---|---|---|---|---|---|---|---|---|---|---|---|---|---|---|---|---|---|
| | Score | Rank | S. | R. | S. | R. | S. | R. | S. | R. | S. | R. | S. | R. | S. | R. | S. | R. |
| PYTORCH DISTR. SHAMPOO | 0.78 | 1 | 0.75 | 1 | 0.75 | 1 | 0.89 | 1 | 0.75 | 1 | 0.75 | 1 | 0.75 | 1 | 0.77 | 2 | 0.81 | 1 |
| SCHEDULE FREE ADAMW | 0.71 | 2 | 0.67 | 2 | 0.67 | 2 | 0.81 | 2 | 0.68 | 2 | 0.68 | 3 | 0.68 | 2 | 0.81 | 1 | 0.67 | 2 |
| GENERALIZED ADAM | 0.64 | 3 | 0.60 | 3 | 0.61 | 4 | 0.59 | 6 | 0.63 | 3 | 0.73 | 2 | 0.59 | 3 | 0.73 | 3 | 0.63 | 3 |
| CYCLIC LR | 0.63 | 4 | 0.58 | 4 | 0.62 | 3 | 0.72 | 3 | 0.62 | 4 | 0.59 | 4 | 0.59 | 4 | 0.72 | 4 | 0.60 | 4 |
| NADAMP | 0.59 | 5 | 0.54 | 6 | 0.57 | 5 | 0.68 | 4 | 0.58 | 5 | 0.55 | 5 | 0.53 | 5 | 0.68 | 5 | 0.60 | 5 |
| BASELINE | 0.57 | 6 | 0.53 | 8 | 0.55 | 6 | 0.65 | 5 | 0.56 | 6 | 0.52 | 7 | 0.52 | 6 | 0.65 | 6 | 0.58 | 6 |
| AMOS | 0.49 | 7 | 0.56 | 5 | 0.50 | 7 | 0.56 | 7 | 0.44 | 7 | 0.42 | 9 | 0.42 | 8 | 0.56 | 7 | 0.47 | 8 |
| CASPR ADAPTIVE | 0.47 | 8 | 0.54 | 7 | 0.40 | 8 | 0.54 | 8 | 0.41 | 8 | 0.54 | 6 | 0.41 | 9 | 0.40 | 8 | 0.54 | 7 |
| LAWA QUEUE | 0.37 | 9 | 0.42 | 9 | 0.32 | 9 | 0.42 | 9 | 0.31 | 9 | 0.42 | 8 | 0.42 | 7 | 0.33 | 10 | 0.31 | 10 |
| LAWA EMA | 0.34 | 10 | 0.25 | 10 | 0.31 | 10 | 0.39 | 10 | 0.28 | 10 | 0.39 | 10 | 0.39 | 10 | 0.39 | 9 | 0.32 | 9 |
| SCHEDULE FREE PRODIGY | 0.00 | 11 | 0.00 | 11 | 0.00 | 11 | 0.00 | 11 | 0.00 | 11 | 0.00 | 11 | 0.00 | 11 | 0.00 | 11 | 0.00 | 11 |

**(b) Self-tuning ruleset**

| | Full | | CRITEO 1TB | | FASTMRI | | RESNET | | VIT | | CON-FORMER | | DEEP SPEECH | | OGBG | | WMT | |
|---|---|---|---|---|---|---|---|---|---|---|---|---|---|---|---|---|---|---|
| | Score | Rank | S. | R. | S. | R. | S. | R. | S. | R. | S. | R. | S. | R. | S. | R. | S. | R. |
| SCHEDULE FREE ADAMW | 0.85 | 1 | 0.83 | 1 | 0.83 | 1 | 0.98 | 1 | 0.83 | 1 | 0.83 | 1 | 0.85 | 1 | 0.83 | 1 | 0.84 | 1 |
| BASELINE | 0.82 | 2 | 0.79 | 2 | 0.82 | 2 | 0.94 | 2 | 0.81 | 2 | 0.79 | 2 | 0.79 | 2 | 0.81 | 2 | 0.79 | 2 |
| NADAMW SEQUENTIAL | 0.33 | 3 | 0.38 | 3 | 0.27 | 3 | 0.38 | 3 | 0.30 | 3 | 0.38 | 3 | 0.29 | 3 | 0.27 | 3 | 0.38 | 3 |
| SINV6 75 | 0.14 | 4 | 0.16 | 4 | 0.12 | 4 | 0.16 | 4 | 0.16 | 4 | 0.16 | 4 | 0.13 | 4 | 0.16 | 4 | 0.08 | 4 |
| SINV6 | 0.09 | 5 | 0.10 | 5 | 0.07 | 5 | 0.10 | 5 | 0.10 | 5 | 0.10 | 5 | 0.09 | 5 | 0.10 | 5 | 0.04 | 5 |
| ADAMG | 0.00 | 6 | 0.00 | 6 | 0.00 | 6 | 0.00 | 6 | 0.00 | 6 | 0.00 | 6 | 0.00 | 6 | 0.00 | 6 | 0.00 | 6 |

A.6    ALGOPERF WORKLOAD WALL-CLOCK TIME COMPARISON BETWEEN JAX & PYTORCH

To measure the wall-clock time performance of JAX & PYTORCH workload implementations, we estimate the submission times for an ADAMW baseline training algorithm to train to target for each of the workloads on the 8×NVIDIA V100 GPUs competition system. The workloads support the same training batch sizes for this ADAMW baseline across frameworks.

The submission time accumulates the wall-clock times of the `update_params` and the `data_-selection` calls and excludes any time spent on logging and checkpointing (which is disabled during scoring runs anyway). During the first few steps of training, there may be some additional overhead in these calls resulting from cache warm-ups and compilation costs of the model and update code. An accurate estimate would be based on enough steps that any especially slow initial steps play only a small role in the average step time. We found that running for 20% of the step hint allowed us to estimate the equilibrium step time well. We then calculated the full submission time by extrapolating the submission time over the 20% step hint to the full step hint.

We initially performed this measurement when the workloads were complete in the sense that they were functionally equivalent. After upgrading the JAX & PYTORCH packages, changing the CUDA driver versions for the hardware configuration, and implementing various improvements and best practices, we then repeated the measurement to capture the final state of the workloads performance across frameworks. The projected submission times of the JAX & PYTORCH workload are presented in Table 8.

**Table 8: Projected submission times for ADAMW baseline on JAX & PYTORCH workloads before and after workload performance adjustments.** All submission times are in minutes. A positive difference indicates that our JAX implementation is faster than our PYTORCH implementation.

| Workload | JAX | | PYTORCH | | Difference | |
|---|---|---|---|---|---|---|
| | Before | After | Before | After | Before | After |
| CRITEO 1TB | 127 | 136 | 213 | 122 | 68% | −10% |
| FASTMRI | 148 | 145 | 163 | 163 | 10% | 12% |
| RESNET | 1047 | 1063 | 1174 | 1135 | 12% | 7% |
| VIT | 1290 | 1378 | 1260 | 1253 | −2% | −9% |
| CONFORMER | 1689 | 1445 | 1755 | 1407 | 4% | −3% |
| DEEPSPEECH | 1047 | 875 | 1229 | 967 | 17% | 10% |
| OGBG | 306 | 399 | 398 | 445 | 30% | 12% |
| WMT | 804 | 782 | 972 | 792 | 21% | 1% |

