# OpenReview forum: "Accelerating neural network training: An analysis of the AlgoPerf competition"
_ICLR.cc/2025/Conference — ICLR 2025 Poster_

### Official Review · Reviewer_XF5p · 2024-10-17

**Soundness:** 2
**Presentation:** 3
**Contribution:** 2
**Rating:** 6
**Confidence:** 1

**Summary:**

This paper investigates the AlgoPerf competition and provides insights into model training acceleration. Some findings are reasonable and could potentially guide the design of training algorithms. As this is a survey rather than a technical paper, I am not sure whether it is qualified to be published in ICLR. Therefore, I seek to consider the opinions of other reviewers.

**Strengths:**

Some interesting findings are provided and they may help to design more efficient training algorithms.

**Weaknesses:**

I am not an expert in evaluating this kind of survey paper, and I am curious about how the findings can be applied to refine existing algorithms.

**Questions:**

Please refer to the weakness part.

---

> ### Author Response · Authors · 2024-11-21
> **Response to reviewer XF5p**
>
> Dear Reviewer XF5p,
>
> Thank you for taking the time to review our paper.
>
> Our paper is not a literature review: it contains novel experimental results, produced by us, that are not published elsewhere. Although we use community-driven submissions to incentivize strong baselines, we ran all the scoring experiments and conducted all the analyses seen in the paper figures. Our paper is akin to a benchmark or meta-analysis, in the tradition of many recent papers accepted to ICLR/ICML/NeurIPS (e.g. [1] from ICML 2021, which shares similar goals).
>
> By thoroughly testing training algorithms, we can determine the state-of-the-art training methods for neural nets, identify which methods truly speed up training, and provide "a valuable focal point to the community for driving future progress in training algorithms" (Reviewer 3MCC).
>
>
> > "I am curious about how the findings can be applied to refine existing algorithms."
>
> Our paper highlights several promising avenues. One key insight is the importance of robustness across workloads.  For example, the Generalized Adam submission successfully trains the ResNet workload, while most other submissions fail to train this workload to the target. Its hyperparameters could inform improvements to Schedule Free AdamW, perhaps by sequentially running it.
> Moreover, our findings pave the way for combining well-performing algorithms into even more efficient methods, such as a hypothetical "Schedule-Free Shampoo." The open-sourced results, including training logs, offer competitive baselines and well-tuned hyperparameters, serving as valuable starting points for further research.
>
>
> [1] Schmidt et al., "Descending through a Crowded Valley - Benchmarking Deep Learning Optimizers", ICML 2021.

---

### Official Review · Reviewer_QEMB · 2024-11-03

**Soundness:** 3
**Presentation:** 3
**Contribution:** 3
**Rating:** 8
**Confidence:** 3

**Summary:**

The paper describes the methodology and results of the "AlgoPerf: Training Algorithms" competition, which aims to evaluate the speed-ups of neural network training by modifying the underlying training algorithm.

The competition covers two rulesets: "external tuning", which requires a hyperparameter search space that is workload agnostic, and "self-tuning", which is hyperparameter-free. This paper details the winners of both rulesets "Distributed Shampoo" and "Schedule Free AdamW".

Finally, the authors detail the issues they encountered while developing this competition, highlighting compatibility and performance issues between different frameworks and improving the respective implementation by copying the better-performing one.

**Strengths:**

S1: Ensuring a fair comparison with JAX and PyTorch is likely impossible, but the authors did a good job of tracing most issues (mathematical correctness, comparing kernel runtimes, etc.). They stopped at memory allocation patterns, which is arguably impossible to get outside of creating an intermediate translation layer between NVIDIA GPU drivers and whatever TPUs are using. This decision showed improvement potential in both frameworks, as the direct comparison between them showcased performance gaps that could be easily closed by copying the better-performing implementation.

S2: The lessons learned are very interesting for practitioners and future competition creators, outlining gaps in current algorithmic development and the dependency on hyperparameter tuning to get the best results.

**Weaknesses:**

W1: I am missing a more detailed analysis of why the winners of the respective rulesets came first. While this paper is more about the competition itself, I would love for it to be slightly more useful for practitioners questioning whether they should drop AdamW for Distributed Shampoo in their experiments. Other questions like whether the current on-trend LLM training will see significant changes due to the results from AlgoPerf (due to the significant cost to training these models) might provide a slightly better outlook and highlight the impact of this competition.

Minor issues:
- Typo in Line 382: "framekworks"

**Questions:**

I would like the authors to address W1.

---

> ### Author Response · Authors · 2024-11-21
> **Response to reviewer QEMB**
>
> Dear Reviewer QEMB,
>
>
> Thank you for your thoughtful and encouraging review. We are happy to hear that you appreciate the efforts on the engineering side.
>
>
> **Regarding "a more detailed analysis of why the winners [...] came first"**
>
> We fully agree that understanding more of why certain methods work is interesting and important. We approached this question mainly by ablating the benchmark decisions, i.e. "Which aspects of the benchmark lead to the winner being ranked first?" For example, we tested the impact on the benchmark scores and ranking when removing individual workloads (Table 8, Appendix) or groups of workloads (Figures 4 & 5, Appendix).  Would you find it valuable to move some of these results to the main text?
>
> Investigating why winners succeed from an algorithmic perspective is a much more complex challenge and deserves a (series of) paper(s) on its own. For instance, this entire paper [1] is dedicated to understanding the gap between Adam & SGD (perhaps the most studied optimizers in deep learning) on one specific model type (Transformers). We would love to tackle this challenging question in future work, ideally in collaboration with the relevant submission authors.
>
> That said, there are a few comments we can make about what made the winning submissions comparatively so effective. As mentioned in the paper, robustness to different workloads was a major factor for the winning submissions. Additionally, implementation quality and efficiency also played a significant role. For non-diagonal preconditioning methods, such as submissions based on Distributed Shampoo or CASPR, creating a time and memory efficient implementation without major bugs is far from trivial.
> Now that AlgoPerf has identified well-performing training methods, we and the entire community can focus on researching and understanding them. Nevertheless, we also believe that already today, the results of the competition provide concrete and practical advice for practitioners. Both Distributed Shampoo and Schedule-Free are great replacements for (traditional) AdamW. The results analysis can also serve as a kind of "lookup table" where practitioners can find results on the benchmark workload closest to their own problem of interest, i.e. not using Schedule Free AdamW for ImageNet ResNet types of workloads.
>
> **Relevance to LLM training**
>
> Unfortunately, most LLM training recipes are proprietary, so it is hard to determine exactly how the models behind the most popular LLM products are trained. However, we are seeing hints (on social media and reading between the lines in papers) of a renewed interest in Shampoo and its application for LLMs over the last few months, indicating that methods that perform well on AlgoPerf might also perform well on large scale language modeling workloads. For some more concrete evidence, this year's ICLR has 4 submissions with Shampoo in the title or the abstract. The paper on SOAP [2], a Shampoo variant, claims a roughly 35% wall-clock speedup over AdamW on LLM pre-training (360M and 660M models). The recent "nanoGPT speedrun" effort also uses Shampoo and Shampoo-inspired methods and is getting promising results consistent with AlgoPerf's results.
>
>
> Finally, thank you for pointing out the typo on line 382. We have corrected it in the updated version.
>
>
> Thank you again for your careful review and insightful comments.
>
> [1] Kunstner et al., "Heavy-Tailed Class Imbalance and Why Adam Outperforms Gradient Descent on Language Models", arXiv 2402.19449, 2024. \
> [2] Vyas et al., “SOAP Improving and Stabilizing Shampoo in Adam”, Under Submission at ICLR 2025.

---

> > ### Comment · Reviewer_QEMB · 2024-11-25
> > **Response to the authors by QEMB**
> >
> > Dear authors,
> >
> > Thank you for responding to my concerns!
> >
> > In W1 I specifically was interested on the algorithmic innovations rather than the performance on held-out workloads. I agree that evaluating this with the needed depth is way outside of the scope of this paper.
> >
> > I would be happy if you could highlight this future line of research with the research community more concretely, specifically under the scope of these kinds of competitions.
> > Maybe something along the lines of "while this competition is broadly useful to determine potential alternatives to well-known optimizers, more detailed analysis like in [1] is needed to understand why they perform better".

---

> > > ### Author Response · Authors · 2024-11-26
> > > **Response to Reviewer QEMB**
> > >
> > > Dear Reviewer QEMB,
> > >
> > > We will try to highlight this future line of research in the paper and update the PDF as soon as we can.
> > >
> > > Thank you again for your suggestions.

---

### Official Review · Reviewer_9ctZ · 2024-11-04

**Soundness:** 2
**Presentation:** 2
**Contribution:** 2
**Rating:** 6
**Confidence:** 2

**Summary:**

This paper analyses the results of the AlgoPerf competition. For that, it presents a summary of the methodology, a detailed description of the winning submissions, how the evaluation was carried out, the implementation details of the competition itself, and the engineering challenges they faced. The main goal of the competition is to evaluate the effectiveness of training algorithms. This is done by measuring how long submissions take to achieve some evaluation goal on some defined workload with restricted runtime and budget. In general, results highlight the competitiveness of the benchmark, as few of the submissions were able to do well on all the different workloads, indicating lots of room for improvement.

**Strengths:**

- Authors provide a detailed analysis of their competition, including the methodology, the best results, and lessons learnt.
 - Originality of the work lies in having the initiative to setup the competition (organisation, dissemination, infrastructure set up), and reporting the results obtained.
 - Analysis is extensive. Authors provide tables and graphics to showcase the results of the competition.
 - Authors provide low level details and lessons learnt also on the implementation and mainteinance of the benchmark, comparing Pytorch and JAX.

**Weaknesses:**

- Novelty: Besides the competition results and the insights obtained, novelty is not high. Contributions are mainly the insights extracted from the submissions. It feels more like a report (summarising results obtained from a competition). I would encourage authors to further highlight the contributions they make, clearly stating that this benchmark is solving a gap, and backing up the claims. In addition, I believe that some of the lessons learnt highlighted in bold are not novel but already established practises (e.g. "having fair comparisons in a competition" is something widely known and established).
 - Clarity: Narrative can be improved in some sections. E.g. Section 3 is specially dense to read, and is not always clear what the authors want to convey. I encourage authors to include, at the start of each paragraph in section 3, a sentence that summarises the main findings of that paragraph. (Eg. ResNet workload subsection. Then, main takaway sentence. Then, the rest of the details, numbers, statistics, etc.)
 - I believe authors could improve the significance of the work by better motivating the need of this benchmark. Why is this benchmark important and needed? Is it the first benchmark to allow evaluation of training algorithms? What makes it different from other benchmarks?  The current paper is lacking a strong motivation background and more evidence. For example, that there is a real need for self-tuning algorithms.

**Questions:**

- "Although a radical change from the current practice of published training algorithms that aren’t runnable without setting various hyperparameters, publishing families of update rules and abdicating responsibility for tuning entirely to the user only adds to the community’s confusion on what to actually use". This is an interesting comment that is hidden in the bulk of the text. It would be interesting if the authors clarified this comment, and expanded on it further.

**Details Of Ethics Concerns:**

No ethical concerns.

---

> ### Author Response · Authors · 2024-11-22
> **Response to reviewer 9ctZ**
>
> Dear Reviewer 9ctZ,
>
> Thank you very much for your detailed and constructive feedback.
>
> **Motivation for the Benchmark and Novelty of this work**
>
> [1] lists well over a hundred neural net training methods (and the list is 3 years out of date, many more have been produced since), most of them published in the last seven years. And yet, despite training methods being such a fundamental part of the deep learning pipeline, the community has been unable to identify which training methods are the best. It is quite difficult to create a convincing, informative, and practically relevant empirical comparison of training algorithms. Without a rigorous and third-party benchmarking process like AlgoPerf, researchers proposing new methods have created their own evaluation setups, which historically have suffered from a number of issues. Most notably, previous empirical comparisons have tended to (1) have weak baselines, (2) not fully account for hyperparameter tuning, and (3) fail to properly control for potential confounding factors, e.g. model architecture changes.
>
> Our work's novelty lies in producing the first competitive comparison of neural network training algorithms that uses a modern, state-of-the-art comparison methodology that properly accounts for hyperparameter tuning, and properly controls for potential confounding factors. For example, Section 4 details all the meticulous engineering work that was necessary for such a fair and informative comparison. This work allows us to identify an almost 30% speedup in neural net training and thus a significant contribution to the community. Among other insights, it convincingly demonstrates that training methods using non-diagonal preconditioners can be faster in wall-clock runtime than the currently dominating diagonal methods, such as Adam.
>
> We will try to revise the text to make the motivation clearer and give a better summary of the motivation for the AlgoPerf methodology from Dahl et al. that we are building on.
>
> **Clarity**
>
> Thank you for your suggestions on how to make Section 3 clearer. We will add summary sentences to each paragraph in this section to better highlight the key insights.
>
> **Your Question**
>
> Currently, a big practical issue in the usability of training methods is that many choices are still left to the practitioner. E.g. how should the learning rate be tuned? In what range? Using what schedule? These are very crucial decisions that can make or break the training process and—critically—determine which methods perform best. By adding the hyperparameter search space to the submission, the community gets a precise recipe for using these methods in practice, including all the necessary details.
> We will try to revise the text to make this point more salient and would welcome any specific suggestions on how to do that.
> Thanks again for your constructive comments and feedback.
>
> [1] Schmidt et al., "Descending through a Crowded Valley - Benchmarking Deep Learning Optimizers", ICML 2021.

---

> > ### Comment · Reviewer_9ctZ · 2024-11-22
> > **Response to Authors**
> >
> > Dear authors,
> >
> > Thanks for your detailed response.
> >
> > **Motivation for the Benchmark and Novelty of this work**
> >
> > This is exactly the kind of motivation I was lacking in the paper. Thank you. Conveying a clear motivation is key, and I think that the two paragraphs authors have provided in their response explaining the motivation are great. (It is important to note that, generally, authors are very familiar with the topic and perfectly know the motivations for the work they do. But a reader might not be too familiar with the specifics, thus, it is key to state clearly the motivation for the paper.) Thanks for addressing that. And I would encourage authors to convey this suggestion into their paper.
> >
> > **"Your Question"**
> >
> > Similarly, the paragraph authors provide in the review ("Currently, a big practical issue in the usability of the training...") is very enlightening. I strongly encourage authors to make this points clear in the paper at the start. Pointing out these reasons at the start of the paper provides a strong narrative, and motivates the rest of the work (ie. now the reader understands why the benchmark is important and why should he care about it). Finally, using simple terms and practical examples (like authors did in the review comment: "how should the learning rate be tuned?...") helps enormously to understand.

---

> ### Author Response · Authors · 2024-11-26
> **Response to Reviewer 9ctZ**
>
> Dear Reviewer 9ctZ,
>
> Thank you again for your suggestions. We will try to integrate the above paragraphs into our paper. However, it will take us some time to weave them seamlessly into the current text and ensure that the main text remains within 10 pages. We will update the PDF as soon as possible.
>
> We also kindly request to consider updating your score if this addresses your concerns.

---

### Official Review · Reviewer_Ed5X · 2024-11-04

**Soundness:** 3
**Presentation:** 4
**Contribution:** 2
**Rating:** 5
**Confidence:** 4

**Summary:**

This paper details the experience of the "AlgoPerf Competition: Training Algorithms". The goal of the competition is to evaluate neural network training speeds by improving the training algorithms. The competition evaluated submissions under two rulesets: external tuning (using predefined hyperparameter search spaces) and self-tuning (completely hyperparameter-free). The competition also demonstrated that the top-scoring algorithms generalized across workloads. For the former, distributed shampoo outperformed other techniques, and for the latter, Schedule Free AdamW demonstrated superior performance.

The paper also describes future training algorithm developments -- emphasizing the importance of fair benchmarking, providing complete algorithm specifications, and different hyperparameter tuning strategies. The paper is written like an experience paper, demonstrating methods and techniques that help with neural network speedups, as well as conducting a fair evaluation of different methods.

**Strengths:**

- The papers' winners (Dist. Shampoo, and Adam W) are interesting to note, and offer strong baselines for the workloads used in the paper.
- The paper describes engineering effort needed to bring parity between Jax and Pytorch, which can be useful in understanding   accuracy/performance differences between the two frameworks related to specific features/API calls that were used in the competition.
- The paper details the engineering and compute needed in hosting a systematic model evaluation framework/process.
- The paper is well-written, and describes the methodology, results and lessons clearly.

**Weaknesses:**

- Weak conclusions:  The authors are encouraged to draw stronger conclusions from the experience. While it is acknowledged that these types of papers are difficult to write, the broad applicability or lessons can be difficult to grasp for the reviewer. The specific nuances in performance evaluation is interesting. But can these results be made more general or useful to improve the paper?  E.g. can you claim that Pytorch/JAX parity is impossible to achieve for specific workloads?
- Unclear fit with ICLR: The paper reads more like an experience report (e.g. Kaggle summaries), rather than a research paper. While the experiences are interesting, the novel contributions/lessons are limited. The lack of a test set and lack of common workloads also limit the applicability. The paper would be likely be a better fit for a software engineering conference both in terms of fit and conference attendee interests.
- Challenges with methodology: The competition evaluation is resource intensive and uses a validation test. Most competitions are evaluated on test sets, and a note describing how the results/methodology can be extended to include test sets would help improve the paper.

**Questions:**

1) Please describe a fit with ICLR, and how publishing this paper helps the broader ICLR community.
2) Can you please provide more insights into the specific reasons behind the significant compute costs, and are there suggestions for optimizing the evaluation process without compromising the robustness of the benchmark?
3) Can you comment on increase in compute costs if test sets and LLMs are considered for model evaluation?

---

> ### Author Response · Authors · 2024-11-21
> **Response to reviewer Ed5x (part 1)**
>
> Dear Reviewer Ed5X,
>
> Thank you for your detailed feedback. We’re happy to address your questions (Q) and comment on the perceived weaknesses (W) you raised.
>
> **Fit with ICLR (Q1, W2)**
>
> ICLR's Call for Paper explicitly mentions "datasets and benchmarks" and "infrastructure, software libraries, hardware, etc."
> Our paper is directly relevant to neural network training, which is at the very heart of—and of critical importance to—the ICLR community. AlgoPerf's winners demonstrate reliably faster training, cutting down training time and compute costs. Researchers studying or developing new training methods will benefit by having strong baselines (along with hyperparameters) to compare to. Our analysis can provide a signal for promising directions for future research in training algorithms. Furthermore, researchers who will make their own training algorithm comparisons or benchmarks will benefit from our experience, in particular, our engineering section (Section 4). Our paper also provides best practices useful for anyone wanting to optimize their code for efficiency in the JAX and PyTorch frameworks.
>
> Moreover, prior work with a similar spirit has been published at ICLR and related conferences (e.g., [1-5]). Feedback from Reviewers 3MCC and QEMB further demonstrates significant interest in our work within the ICLR community.
>
> [1] Bai et al., "Benchmarking Algorithms for Federated Domain Generalization," ICLR 2024 (spotlight). \
> [2] Yu et al., "KoLA: Carefully Benchmarking World Knowledge of Large Language Models," ICLR 2024. \
> [3] Schmidt et al., "Descending through a Crowded Valley - Benchmarking Deep Learning Optimizers," ICML 2021. \
> [4] Agarwal et al., "Deep Reinforcement Learning at the Edge of the Statistical Precipice," NeurIPS 2021 (outstanding paper). \
> [5] Montali et al., "The Waymo Open Sim Agents Challenge," NeurIPS 2023 (Datasets and Benchmarks Track).
>
> **Broadly applicable lessons (W1)**
>
> Ultimately, we view this work as producing the first competitive comparison of neural network training algorithms that uses a modern, state-of-the-art comparison methodology that properly accounts for hyperparameter tuning and properly controls for potential confounding factors due to workload, framework and hardware details.
> A broadly applicable lesson from our work is that training algorithms cannot be separated from tuning protocols. Therefore papers introducing new training algorithms should publish something actually runnable by providing a tuning protocol along with evidence that it generalizes across workloads (perhaps by evaluating it on the AlgoPerf leaderboard). The community can finally move away from every paper introducing a new training algorithm also introducing a new evaluation protocol.
>
> **Is PyTorch/JAX parity impossible to achieve for specific workloads? (W1)**
>
> PyTorch/JAX parity is a moving target because these frameworks are actively developed frameworks with ever-evolving features and best practices.
> At a given moment in time,  we should view PyTorch/JAX parity as a continuum where we can invest engineering labor to increase parity. Therefore the operative question is not whether parity is possible but whether there is sufficient parity to make meaningful algorithmic comparisons.
>
>
> *(please see part 2 for further response)*

---

> ### Author Response · Authors · 2024-11-21
> **Response to reviewer Ed5X (part 2)**
>
> *(continuation of part 1)*
>
> **Common Workloads (W2)**
>
> The benchmark includes a fully-connected neural network with large embeddings, ResNet, Transformer sequence model, Conformer, Vision Transformer, U-Net, Graph Neural Network, and LSTM. This covers a wide range of popular and widely used model architectures. That said, any finite library of workloads will be limited. Is there a particular model architecture missing that you think we should call out specifically in the limitations section?
>
> **What is the proper role of a test set when benchmarking on known, existing workloads? (W2)**
>
> AlgoPerf relies on existing datasets and cannot access additional, unknown test sets. Since we didn't create these original datasets, we can't collect additional private test sets for them. This limitation means we cannot guarantee that community submissions avoided using the original test sets during algorithm development. Therefore, we decided it is more appropriate to be transparent about this point and clearly mark the held-out data as "validation sets".
> That said, our benchmark employs randomly sampled held-out workloads, mimicking the function of test sets on the workload-level to evaluate generalization to new workload variants. We could easily use the "test sets" for each component dataset associated with the benchmark workloads in the scoring protocol, but this would be more of a terminology change than a methodological one. The primary barrier to overfitting are the various limitations on workload-specific hyperparameter tuning and the requirement for submissions to perform well across the entire pool of workloads, jointly.
>
> **Contributors to Compute Costs (Q2)**
>
> A significant part of the compute costs comes from using workloads of a large enough scale to be practically relevant. We must have a diverse set of such workloads since we care about identifying general-purpose methods that can efficiently train generic neural networks. This is in contrast to workload-specific competitions, such as training ImageNet as fast as possible, which usually result in hyper-specific setups that provide little value to most practitioners with their own workloads.
> We also wanted to train until a competitive performance is reached to ensure meaningful results since methods that excel in reaching weak targets don’t necessarily perform well at achieving competitive ones.
> Lastly, we wanted to ensure that our results are robust, which is why we repeat our process 5 times (called "studies" in our paper). This ensures that the insights are robust and not just a result of random noise due to the stochastic training process, although in hindsight this was probably overkill.
>
> **Ideas for Reducing Compute Costs (Q2)**
>
> We proposed some ideas that will result in cost reductions in section 5.1 (replacing held-out workloads with a smaller number of additional base workloads and reducing runtime budgets), but we can make this text more explicit and add additional suggestions.
> For example, we could (1) reduce the number of studies (repetitions with different seeds for statistical fidelity) from 5 to 3, reducing costs by an additional 40%. (2) use a more modern hardware setup (compared to the 8xV100s) to achieve a better "cost-to-performance-ratio". We are happy to add those considerations to the text of the paper.
>
> **Additional Costs When Considering Test Sets and LLMs (Q3)**
>
> While the cost increases from adding workload-specific test sets (see above for why we can't guarantee that they won't be used during submission development) would be negligible, including truly massive-scale language models is not feasible. If a language model gets added to future benchmark iterations, it would be sized to be near one of the first rungs of a typical "scaling ladder" and smaller than typical production scale. However, a smaller model could still provide valuable signals for algorithmic research. Recently, there have been interesting results in training smaller LMs to increasingly competitive performance and there have been anecdotal reports that some of these insights generalize to larger scales.
>
> We hope our responses have addressed your concerns and clarified the contributions of our work. If so, we kindly ask you to consider updating your evaluation.

---

> ### Comment · Reviewer_Ed5X · 2024-11-27
> **Re:Rebuttal**
>
> In the papers, that you've linked: new research results or datasets are being introduced. However, the current paper does read like a competition report (w/o a held out test set). I agree that this will be a good contribution to the datasets and benchmarks track but currently it is reading more like a Kaggle competition report. I would still encourage the authors to consider drawing stronger * research* conclusions from the experience report.

---

### Official Review · Reviewer_3MCC · 2024-11-05

**Soundness:** 4
**Presentation:** 4
**Contribution:** 4
**Rating:** 10
**Confidence:** 4

**Summary:**

This paper presents an analysis of the results of the recent AlgoPerf Training benchmark, in which a variety of community-submitted algorithms were evaluated on multiple workloads and in multiple settings to identify those which yield improved training algorithms. A variety of details from the benchmark results are presented, leading to some broad trends (e.g., the best optimizers are those that are "consistently reasonable" as no one approach dominated all workloads) as well as suggestions for future directions. The paper also includes lessons learned and commentary on the benchmark itself, and on the engineering efforts involved.

**Strengths:**

The paper summarizes and analyzes the results of the AlgoPerf Training benchmark, providing a valuable focal point to the community for driving future progress in training algorithms and setting the agenda for research. The current advances and limitations of training algorithms are highlighted, helping to clearly identify areas for improvement in the community.

Equally valuable, the paper has a detailed discussion of lessons learned and suggestions from the process of running the competition. These are details that are often not widely disseminated, and are valuable for others seeking to build similar benchmarks. This includes a discussion of engineering challenges involved in ensuring fair and reasonable comparisons across submissions and frameworks.

Overall the paper is clear, well-written, and likely to help drive progress in the ML community.

**Weaknesses:**

I have no notable concerns about the paper.

Very minor typo: L382, "framekworks" -> "frameworks"

**Questions:**

n/a

---

> ### Author Response · Authors · 2024-11-21
> **Response to reviewer 3MCC**
>
> Dear Reviewer 3MCC,
>
> Thank you very much for your positive and very supportive review of our paper and your time and effort. We are especially pleased that you agree that our work will help drive future progress. \
> Thank you for pointing out the typo, we have corrected it in our updated version.
>
> Your review is a huge encouragement for our effort.

---

> > ### Comment · Reviewer_3MCC · 2024-11-25
> >
> > You're very welcome, and I continue to remain positive on this paper.

---

### Comment · Area_Chair_YPFs · 2024-11-22
**Discussion**

Dear reviewers,

The authors have responded to your reviews.

Until November 26th @ 2359 (AOE time) reviewers and authors can freely exchange responses, so if there any clarifications you require from the authors, now is the time to seek them!

Best,

AC

---

### Meta-Review · Area_Chair_YPFs · 2024-12-11

**Metareview:**

This paper provides an analysis of the AlgoPerf competition results, which compares neural network training algorithms across different workloads with, and without hyperparameter tuning. The findings received appreciation from all reviewers. Most of the criticism boiled down to whether this was in scope for ICLR and the question of whether it counted as "research".

It is my opinion that this is clearly in scope, falling under the "datasets and benchmark" topic (https://www.iclr.cc/Conferences/2025/CallForPapers). It is also my opinion that this analysis counts as research.

I agree with Reviewer 9ctZ that this work could be motivated better. It is important stuff. Everyone is training (many) neural networks and generalisable lessons for making this training faster are of huge benefit. I encourage the authors to incorporate the motivation they provided in https://openreview.net/forum?id=CtM5xjRSfm&noteId=CZBzKuLsB3 in the paper introduction.

I believe this should be accepted as a spotlight. Researchers should be aware of this benchmark as it can lead to the development of techniques that reliably train networks faster.

**Additional Comments On Reviewer Discussion:**

I don't believe any reviewers changed their scores, although there were some very high scores to begin with (10,8) so this wasn't too surprising. I think the authors responded well to the reviewers and were able to address their queries. The only real negative identified was a question of scope but I think this is in scope for ICLR (which I note, unlike NeurIPS, does not have a separate datasets and benchmarks track).

---

### Decision · Program_Chairs · 2025-01-22

Accept (Poster)